# Nolz1 expression is required in dopaminergic axon guidance and striatal innervation

Clement Soleilhavoup[1], Marco Travaglio [1,6], Kieran Patrick[1,6], Pedro Garção[1], Elangovan Boobalan[2], Youri Adolfs[3], Ruth V. Spriggs[1], Emma Moles-Garcia[1], Dalbir Dhiraj[1], Tony Oosterveen[1], Sarah L. Ferri[4], Ted Abel [4], Edward S. Brodkin[5], R. Jeroen Pasterkamp [3], Brian P. Brooks[2] & Lia Panman [1]✉

Midbrain dopaminergic (DA) axons make long longitudinal projections towards the striatum. Despite the importance of DA striatal innervation, processes involved in establishment of DA axonal connectivity remain largely unknown. Here we demonstrate a striatal-specific requirement of transcriptional regulator Nolz1 in establishing DA circuitry formation. DA projections are misguided and fail to innervate the striatum in both constitutive and striatal-specific *Nolz1* mutant embryos. The lack of striatal Nolz1 expression results in nigral to pallidal lineage conversion of striatal projection neuron subtypes. This lineage switch alters the composition of secreted factors influencing DA axonal tract formation and renders the striatum non-permissive for dopaminergic and other forebrain tracts. Furthermore, transcriptomic analysis of wild-type and *Nolz1*$^{-/-}$ mutant striatal tissue led to the identification of several secreted factors that underlie the observed guidance defects and proteins that promote DA axonal outgrowth. Together, our data demonstrate the involvement of the striatum in orchestrating dopaminergic circuitry formation.

[1] MRC Toxicology Unit, University of Cambridge, Hodgkin Building, Lancaster Road, Leicester LE1 9HN, UK. [2] Ophthalmic Genetics & Visual Function Branch, National Eye Institute, National Institutes of Health, Bethesda, MD 20892, USA. [3] Department of Translational Neuroscience, UMC Utrecht Brain Center, University Medical Center Utrecht, Utrecht University, 3584 CG Utrecht, The Netherlands. [4] Department of Neuroscience and Pharmacology, Iowa Neuroscience Institute, University of Iowa, Iowa City, IA 52242, USA. [5] Center for Neurobiology and Behavior, Department of Psychiatry, Perelman School of Medicine at the University of Pennsylvania, Philadelphia, PA 19104-3403, USA. [6] These authors contributed equally: Marco Travaglio, Kieran Patrick. ✉email: liapanman1@gmail.com

Midbrain dopaminergic (DA) neurons play an important role in several brain functions including locomotion, motivation and reward processes[1]. The guidance of DA axons towards their target areas is an important step in the establishment of functional circuits that are required for executing those roles. The projections of several molecularly distinct subpopulations of DA neurons have been defined[2–4] with substantia nigra (SN) and VTA DA neurons innervating the dorsal and ventral part of the striatum respectively, consistent with their specific roles. While the specification of DA neurons and their projection areas have been intensively studied, it remains relatively unclear how the establishment of DA axonal projections and striatal innervation are regulated during embryonic development.

After exiting the midbrain DA axons are attracted by and fasciculated within the medial forebrain bundle (MFB)[5], which forms two rostrally oriented ipsilateral tracts within the ventral diencephalon. These axonal tracts run parallel to the ventral midline towards target areas in the forebrain including the striatum and cortex[6]. Several secreted guidance molecules with either attractive or repulsive activities are involved in the navigation of DA axons towards their target areas, including Slit/Robo, Netrin/Dcc, Ephrin, Semaphorin and Wnt signalling components[7]. Besides extrinsic factors secreted in the environment, axons can also be guided to the target areas by reciprocal axon–axon interactions as demonstrated for the innervation of the lateral habenula[8] and the establishment of thalamocortical–corticothalamic interactions[9]. However, whether establishment of DA connectivity is influenced by striatal patterning or the formation of striatal axonal extensions has not been determined.

The majority of the neurons in striatum are projection neurons, which can be subdivided into two subpopulations based on their transcriptional profile and target innervation[10–12]. Striatonigral projection neurons are specified by several transcription factors including Isl1, Ebf1 and Rarb and directly innervate the SN[13–17]. In contrast, striatopallidal neurons project to the GP and give rise to the indirect pathway. Here we investigate whether and how striatal patterning influences the guidance and target innervation of DA axons.

Nolz1 is as a transcriptional regulator expressed in the VTA DA neuronal lineage and striatal projection neurons[18–21]. Here we show that in the absence of striatal Nolz1 expression DA axons are misguided and fail to innervate the striatum. We demonstrate that the striatonigral to -pallidal switch in projection neuron subtype identity in $Nolz1^{-/-}$ mutant embryos is associated with defects in establishment of DA and forebrain axonal tracts. The altered composition of guidance factors secreted from $Nolz1^{-/-}$ mutant striatum provide a non-permissive environment for DA axons and other forebrain axonal tracts. Transcriptomic analysis resulted in the identification of proteins that can rescue the defects in DA axonal outgrowth. Finally, the acquired insight into mechanisms involved in DA circuitry formation will facilitate the development of approaches to improve graft outcome in cell transplantation studies.

## Results

**Nolz1 is required for establishment DA axonal connectivity.** Previously, we have shown that Nolz1 is expressed in the VTA DA neuronal lineage[18]. To investigate the role of Nolz1 in DA neuron development, we analyzed tyrosine hydroxylase (TH) expression by iDISCO[22] in E18.5 $Nolz1^{-/-}$ mutant embryos, in which the coding region of $Nolz1$ has been replaced by beta-Galactosidase (also referred to as $Nolz1^{bgal/bgal}$) (Supplementary Fig. 1a). IDISCO analysis revealed that DA axons are misguided

in $Nolz1^{-/-}$ mutant embryos (Fig. 1a–d). While in wild-type embryos DA axons extend rostrally through the hypothalamus and innervate the striatum at E18.5 (Fig. 1a, c), a large proportion of TH labelled axons cross the midline in the hypothalamus (arrows in Fig. 1b, d) and terminate rostral of the striatum (arrowheads in Fig. 1b, d) in the mutant embryos. Analysis of NOLZ1 expression in relation to DA axons labelled by GlycoDAT and TH shows that NOLZ1 is expressed in regions that display the guidance phenotype e.g. the hypothalamus (arrow Fig. 1e, q, s) and striatum (arrowhead Fig. 1g, i)[19–21]. It further confirmed that a subset of DA axons cross the midline (Fig. 1e, f, k, l, q, r) and the remaining axons terminate rostral of the striatum (Fig. 1i, j, u, v) resulting in a lack of innervation of the rostral areas (Fig. 1g, h, m, n).

To investigate whether the DA axons that cross the midline have a different identity in comparison with the axons that keep following their trajectory towards the striatum, we analyzed the expression of GlycoDAT, which is higher expressed in SN neurons and their projections compared with VTA neurons[23]. Interestingly, mainly GlycoDAT+TH+ SN DA axon bundles crosses the midline (Fig. 1k, l), while GlycoDAT−TH+ axons extend towards the striatum (Fig. 1m, n), indicating that SN and VTA derived DA axons respond differentially to the absence of Nolz1 expression. Furthermore, the DA axon bundle is defasciculated and more spread out along the medial–lateral axis in $Nolz1^{-/-}$ mutant embryos (Fig. 1o, p). Other axonal tracts running in the MFB are also misguided with 5HT labelled serotonergic axons crossing the midline alongside DA axons (Supplementary Fig. 1b).

While the caudal part of the striatum remains innervated to a certain extent, the globus pallidum (GP) is totally devoid of any DA terminals in $Nolz1^{-/-}$ mutant embryos (Fig. 1i, j, w, x and Supplementary Fig. 2a). The GP specification is normal as shown by CTIP2 (Fig. 2h, j) and NKX2.1 (Supplementary Fig. 2b) expression in $Nolz1^{-/-}$ mutant embryos. However, in the absence of Nolz1 we observed several aberrantly located ISL1+ positive cells (Supplementary Fig. 2c) within in the GP, which could underlie the defect in striatal innervation by DA axons[24].

Finally, the axon guidance phenotype observed in embryos lacking Nolz1 expression was not caused by the mislocalization or elimination of mutant cells as bGAL expression in $Nolz1^{bGal/bGal}$ mutant embryos was similar to Wt and heterozygous embryos in the midbrain (Supplementary Fig. 1c), hypothalamus (Fig. 1q–t (arrows)) and the striatum (Figs. 1u–x, 2w).

**Nolz1 is required for striatal projection neuron development.** The DA axon guidance phenotype observed in $Nolz1^{-/-}$ mutant embryos could not be explained by defects in DA neuron differentiation or hypothalamic patterning as no striking changes in DA neuron and hypothalamic marker expression could be observed (Supplementary Figs. 1c–e, 3a, b and Supplementary Table 1). However, we found that Nolz1 is required for the specification of striatal projection neurons. NOLZ1 is expressed by all striatal projection neurons as demonstrated by its overlap with DARPP32 (Fig. 2a, a′) and CTIP2 (Fig. 2g, g′) expression[20,21]. In $Nolz1^{bgal/bgal}$ mutant embryos striatal projection neurons, labelled by CTIP2 and bGAL, lack DARPP32, indicating that there is a defect in their maturation (Fig. 2b, b′, d, d′). DARPP32 expression appears in clusters in $Nolz1^{-/-}$ mutant striatum that are positively labelled by L1 (Fig. 2f), indicating that these clusters represent abnormally fasciculated axonal tracts. There was a loss of SN innervation by DARPP32+ labelled striatal projection neurons in $Nolz1^{-/-}$ mutant embryos (Fig. 2q–t). In contrast, the expression of CTIP2[25] (Fig. 2g, g′, i, i′, w) was unchanged in $Nolz1^{bgal/bgal}$ mutant, suggesting that the differentiation of striatal projection

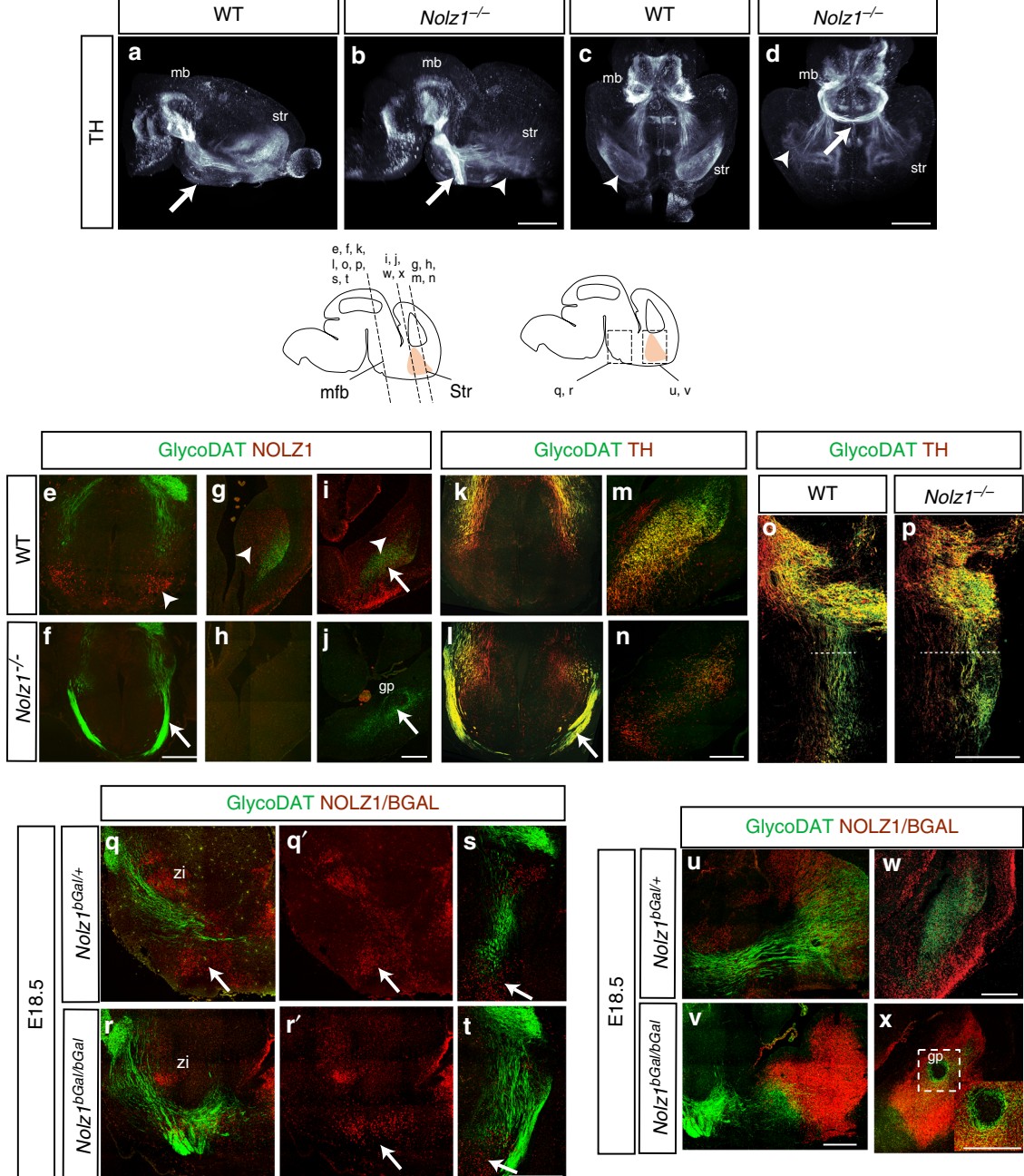

**Fig. 1 *Nolz1⁻/⁻* mutant embryos display DA axon guidance defects. a–d** Visualization of TH⁺ DA axon bundles in E18.5 Wt and *Nolz1⁻/⁻* mutant mouse brains stained and cleared according to the iDISCO protocol. **a** Arrow point to DA axon bundle running in parallel to the ventral midline. **b, d** Arrows indicate midline crossing of DA axon bundle in *Nolz1⁻/⁻* mutant embryos. **b** Arrowhead points to axons terminating in front of the striatum. **c** Arrowhead points to normal DA innervation of striatum in Wt embryos, which is disrupted *Nolz1⁻/⁻* mutant embryos (arrowhead in **d**). Sagittal view in **a, b** and ventral view in **c, d**. **e–j** Immunohistochemical analysis of GlycoDAT and NOLZ1 expression in the hypothalamus (**e, f**) and striatum (**g–j**) of E18.5 mouse Wt and *Nolz1⁻/⁻* mutant embryo. Arrows indicate midline crossing of DA axon bundles in the *Nolz1⁻/⁻* mutant hypothalamus (**f**). **i, j** Arrows point to the GP, which lacks innervation in *Nolz1⁻/⁻* mutant embryos. Arrowheads pointing to NOLZ1 positive cells in the hypothalamus (**e**) and striatum (**g, i**) of Wt embryos. **k–p** Immunostaining showing expression of GlycoDAT and TH in the hypothalamus (**k–l**), striatum (**m–n**) and caudal diencephalon/midbrain (**o–p**) in E18.5 Wt and *Nolz1⁻/⁻* mutant embryos. Arrow in (**l**) indicates GlycoDAT⁺ TH⁺ DA axons crossing the midline in *Nolz1⁻/⁻* mutant hypothalamus. **o, p** Dashed line indicates the width of the DA axon bundle extending from *Nolz1⁻/⁻* mutant midbrain. **q–x** Expression of GlycoDAT and NOLZ1/BGAL in the diencephalon (**q–t**) and striatum (**u–x**) of Wt and *Nolz1⁻/⁻* mutant embryos. Arrows indicating NOLZ1⁺BGAL⁺ labelled cells in *Nolz1bgal/+* heterozygous (**q, q′, s**) and bGAL labelled cells in *Nolz1bGal/bGal* homozygous mutant (**r, r′, t**) hypothalamus. **u, v** DA axon bundles terminate in front of striatum in *Nolz1bGal/bGal* mutant embryos. **w, x** GP in *Nolz1bGal/bGal* mutant embryos is devoid of BGAL labelled cells and lack innervation by DA terminals. **q, r, u, v** sagittal and **s, t, w, x** coronal view. Data are representative of two (**a–d**) or three (**e–x**) independent experiments. Mb (midbrain), Str (striatum), GP (globus pallidus), Zi (Zona Incerta), Hth (hypothalamus). Scale bar 1000 μm (**a–d**); 200 μm (**e–x**).

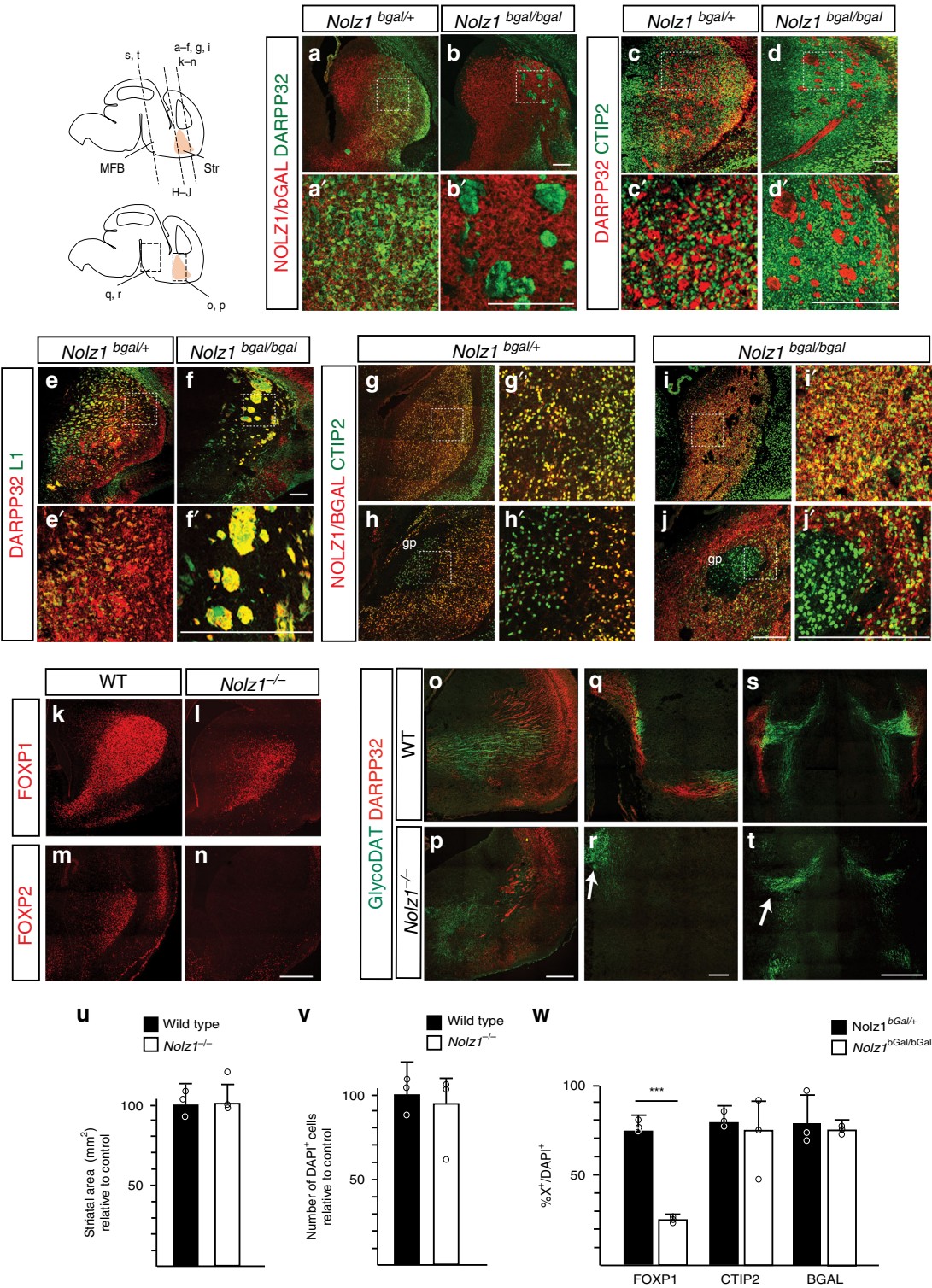

neurons was normally initiated. In addition, striatal projection neuron markers FOXP1 and FOXP2[26,27] were downregulated in *Nolz1*[−/−] mutant striatum (Fig. 2k–n, w). There was no change in cell number, striatal volume and number of bGAL labelled cells between *Nolz1*[bGal/+] and *Nolz1*[bGal/bGal] mutant striatum (Fig. 2u, v, w). Consistently, there was no increase in apoptosis in the absence of striatal Nolz1 expression (Supplementary Fig. 2d). These data show that striatal Nolz1 expression is required for the generation of mature striatal projection neurons and their projections towards the midbrain.

**Differential effect of striatal genes in the absence of Nolz1.** To gain further insight into the transcriptomic changes that underlie the phenotypic alteration in *Nolz1*[−/−] mutant striatum we compared the gene expression profile of E18.5 wild-type with *Nolz1*[−/−] mutant striatum using RNA sequencing (Fig. 3a). The differentially expressed genes (Supplementary data 1) were subjected to hierarchical clustering and displayed in a heatmap, which revealed clusters of genes that were either higher (yellow) or lower (blue) expressed in the wild-type compared with mutant striatum (Fig. 3b). The transcriptomic data confirmed the

**Fig. 2 Striatal projection neurons fail to develop in Nolz1$^{-/-}$ mutant embryos. a–b** Immunostaining showing DARPP32 and NOLZ1/BGAL expression in the striatum of E18.5 Nolz1$^{bgal/+}$ and Nolz1$^{bgal/bgal}$ mutant embryos. **c–d** Immunohistochemical analysis of CTIP2 and DARPP32 staining in Nolz1$^{bgal/+}$ and Nolz1$^{bgal/bgal}$ mutant striatum at E18.5. **e–f** DARPP32/L1 staining in the striatum of E18.5 Nolz1$^{bgal/+}$ and Nolz1$^{bgal/bgal}$ mutant embryos indicate abnormal overlap between L1 and DARPP32 in Nolz1$^{bgal/bgal}$ mutant striatum. **g–j** CTIP2 and NOLZ1/BGAL expression in the rostral (**g, i**) and caudal (**h, j**) striatum of E18.5 Nolz1$^{bgal/+}$ and Nolz1$^{bgal/bgal}$ mutant embryos. Note the absence of NOLZ1/BGAL in the GP of Nolz1$^{bgal/+}$ (**h, h′**) and Nolz1$^{bgal/bgal}$ (**j, j′**) mutant embryos. **k–l** Immunofluorescence analysis of FOXP1 in the striatum of E18.5 Wt and Nolz1$^{-/-}$ mutant striatum. **m–n** Immunostaining showing FOXP2 expression in E18.5 Wt and Nolz1$^{-/-}$ mutant striatum. **o–t** Immunohistochemical analysis of GlycoDAT and DARPP32 expression in E18.5 Wt and Nolz1$^{-/-}$ mutant embryos. (**o, p**) Sagittal sections of striatum showing reduced levels of DARPP32 and GlycoDAT labelled DA terminals in Nolz1$^{-/-}$ mutant embryos. (**q–r**) Sagittal sections of the midbrain/diencephalon. Arrow in (**r**) indicates the absence of DARPP32$^+$ striatal axons projecting towards the midbrain in mutant embryos. **s–t** coronal sections of the midbrain/diencephalon. **t** Arrow indicates the absence of DARPP32$^+$ striatal axons innervating DA neurons in midbrain in Nolz1$^{-/-}$ mutant embryos. **u** Graph shows the relative difference of the striatal area (mm$^2$) between E15.5 Wt and Nolz1$^{-/-}$ mutant embryos (n = 3 biologically independent samples). Wt values are normalized to 100%. Mean values ± standard deviation. **v** Graph shows the relative difference in DAPI$^+$ cell number between E15.5 Wt and Nolz1$^{-/-}$ mutant striatum. (n = 3 biologically independent samples). Wt values are normalized to 100%. Mean values ± standard deviation. **w** Graph indicates the percentage of FOXP1$^+$, CTIP2$^+$ and bGAL$^+$ cells against the total number of DAPI$^+$ cells (n = 3 biologically independent samples) in the striatum of E18.5 Wt and Nolz1$^{-/-}$ mutant embryos. Mean values ± standard deviation. Two-sided, unpaired T-test ***p = 1.86067xE$^{-06}$. Data are representative of three independent experiments (**a–t**). Scale bar: 200 μm (**a–r**), 500 μm (**s–t**). Source data are provided as a source data file.

downregulation of generic striatal projection neuron markers DARPP32, FOXP1 and FOXP2 as observed in Nolz1$^{-/-}$ mutant embryos (Fig. 3f and Supplementary data 1). Interestingly, analysis of the genes representing the different clusters revealed that several genes downregulated in Nolz1$^{-/-}$ mutant embryos (clusters 1, 2, 5b and 5c) are selectively expressed in striatonigral projection neurons, while striatopallidal neuron specific genes were expressed higher in the mutant (clusters 3, 4, 5a and 5e)[10–12]. We at randomly selected 61 differentially expressed genes for further verification and we could validate the expression of 67% of these genes (Supplementary data 2). The striatonigral specific genes, including Drd1, Rarb, Ebf1[11], Isl1, Zfp521, Tac1 and Pdyn were either downregulated or absent in the E15.5 Nolz1$^{-/-}$ mutant striatum (Fig. 3d, f). In addition, striatopallidal neuron specific genes Drd2, Six3, Grg4 (Tle4), Grik3, Penk, Ptprm and Adora2a were upregulated in the mutant striatum, except for Gpr6 (Fig. 3e, f). Despite the upregulation of several pallidal markers, the projections of striatopallidal neurons towards the GP were impaired in Nolz1$^{-/-}$ mutant embryos as revealed by the analysis of PENK (Supplementary Fig. S2e). Genes involved in early striatal patterning, progenitor-, glial- and interneuron specification were not changed in Nolz1$^{-/-}$ mutant striatum (Supplementary data 1). Thus, Nolz1 is selectively required for the specification of striatonigral projection neurons and in the absence of Nolz1 expression several striatopallidal markers are ectopically expressed in the striatum.

**Striatonigral to pallidal switch in Nolz1$^{-/-}$ mutant striatum.** Next, we investigated the specification of the striatal projection neuron subtypes in Nolz1$^{-/-}$ mutant embryos in more detail. In E18.5 Nolz1$^{bgal/+}$ heterozygous mutant embryos about 40% of the striatal projection neurons labelled by bGAL express the striatopallidal marker SIX3 (Fig. 4c), which is expanded to nearly all (90%) striatal projection neurons in Nolz1$^{bgal/bgal}$ mutant embryos (Fig. 4c). Since there is no reduction in total number of striatal projection neurons (Fig. 2v, w), the striatopallidal selective genes are most likely upregulated at the expense of nigral-specific markers in Nolz1$^{-/-}$ mutant embryos. The temporal analysis of striatonigral and pallidal markers at several embryonic stages shows that the nigral to pallidal switch occurs at the time projection neurons are born. The striatopallidal lineage specific markers Six3, Grg4 and Drd2 are upregulated in Nolz1$^{-/-}$ mutant embryos compared with Wt embryos at E11.5 and E12.5 (Fig. 4a, b). Immunolabelling shows that SIX3 is expressed in all bGAL labelled striatal projection neurons in E11.5 Nolz1$^{bgal/bgal}$ mutant embryos, while the expression of striatonigral specific marker

EBF1 (Fig. 4a, b) is not induced at any examined stage. Together, the temporal expression analysis demonstrates that the striatonigral to pallidal lineage switch in Nolz1$^{-/-}$ mutant embryos coincides with the timing of striatal projection neuron production.

During striatal development the timing of cell cycle exit contributes to the subtype identity of striatal projection neurons[28–30] and the majority of neurons born at E11.5 have striatonigral specific identities with the number of striatopallidal neurons increasing over time[29,31,32]. To analyse whether altered timing of striatal projection neuron production contributes to the fate change in Nolz1$^{-/-}$ mutant embryos we administered Bromodeoxyuridine (BrdU) to pregnant females at E10.5, E11.5 or E12.5 and analyzed the embryos at E15.5. While the percentage of BrdU labelled striatal neurons increases between E10.5 and E12.5 in Wt embryos, significantly less neurons were born in E11.5 Nolz1$^{-/-}$ mutant striatum (5% in mutants vs. 20% in wild-type) (Fig. 4d and Supplementary Fig. 4e). No differences in BrdU labelling between Wt and mutant embryos were observed when BrdU was injected at E10.5 or E12.5 (Fig. 4d and Supplementary Fig. 4e). The majority of the neurons labelled by BrdU resulting from injections at E11.5 and E12.5 expresses FOXP1 (Fig. 4e, f), indicating that mainly projection neurons are born at these stages in both Wt and Nolz1$^{-/-}$ mutant embryos. The reduced levels of BrdU labelling at E11.5 in Nolz1$^{-/-}$ mutant embryos coincide with the timepoint when the majority of neurons born in Wt embryos are positive for the striatonigral specific marker EBF1[11] (Supplementary Fig. 4a–c) and is consistent with the absence of striatonigral neurons in Nolz1$^{-/-}$ mutant embryos. In contrast to the Wt striatum, the majority of BrdU labelled neurons in Nolz1$^{-/-}$ mutant striatum expresses the pallidal marker SIX3 (Supplementary Fig. 4f, g). This further demonstrates that neurons born between E10.5 and E12.5 have already been directed towards the striatopallidal fate. Furthermore, there is a strong reduction of the striosomal marker MOR1 (Supplementary Fig. 4h) and an expansion of the matrix marker CALBINDIN1 (Supplementary Fig. 4i) in Nolz1$^{-/-}$ mutant striatum, which is in line with the loss of early born neurons[30,32]. No changes in BrdU labelling was observed in the cortex (Supplementary Fig. 4d, e). The absence of Nolz1 in the ventricular zone and in proliferating cells within the subventricular zone[21] (Supplementary Fig. 4j), suggests that Nolz1 regulates the temporal production of striatal projection neurons non-cell-autonomously through a yet unknown mechanism.

**Striatal selective requirement Nolz1 in DA axon guidance.** To investigate in which brain region Nolz1 function is required for guiding DA axons to the striatum in more detail, we crossed

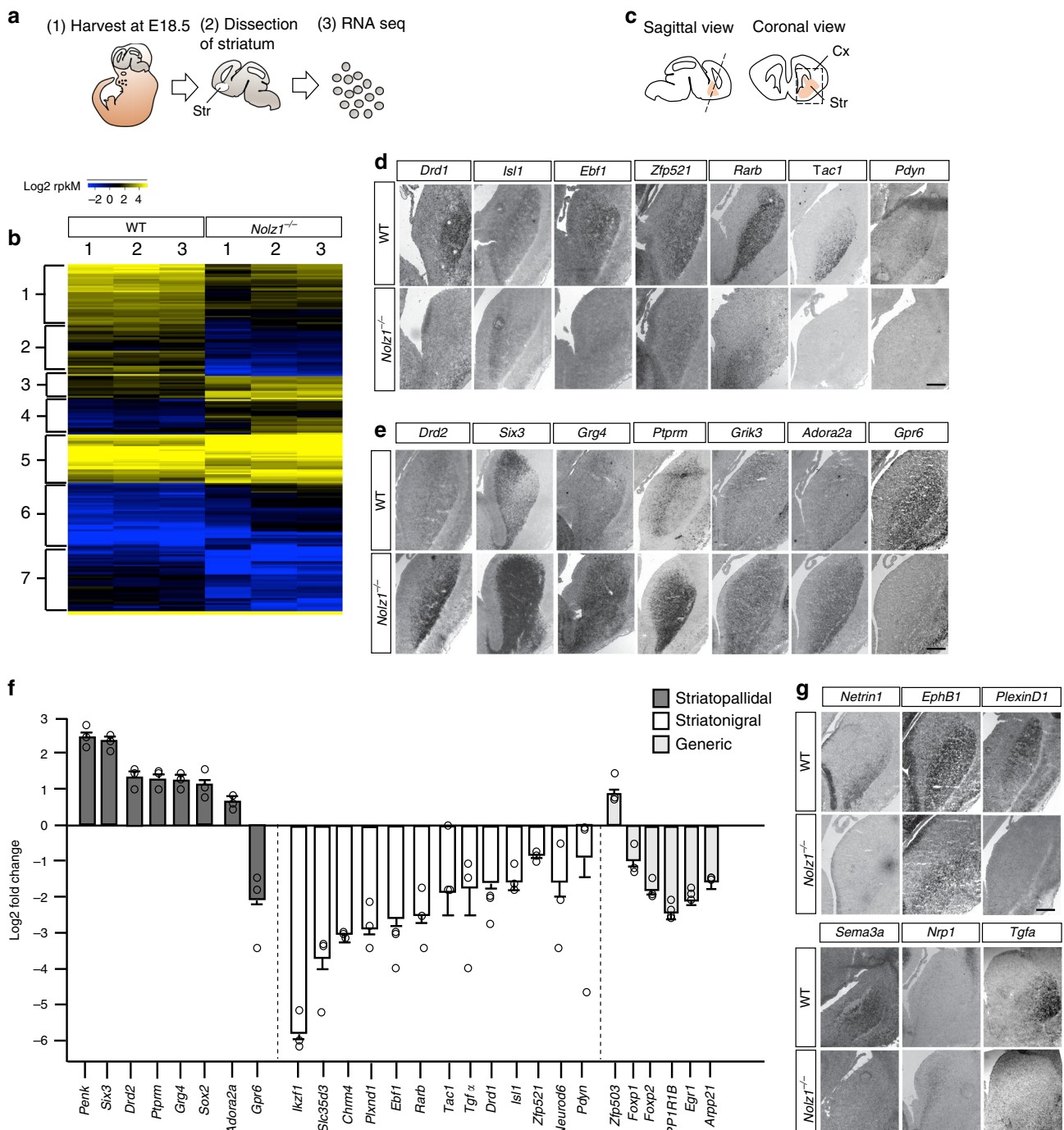

**Fig. 3 Striatonigral to -pallidal switch of projection neuron identity in *Nolz1⁻/⁻* mutant striatum. a** Schematic outline of dissection followed by RNA sequencing analysis of Wt and *Nolz1⁻/⁻* mutant striatal tissue. **b** Heatmap showing differentially expressed genes in the E18.5 Wt and *Nolz1⁻/⁻* mutant striatum ($n = 3$ biologically independent samples). Upregulated genes are shown in yellow and downregulated genes in blue. Hierarchical clustering indicates that clusters 1, 2, 5b and 5c represents striatonigral and clusters 3, 4, 5a and 5e striatopallidal selective genes. **c** Schematic representation of section plane used to obtain coronal sections through the striatum of E15.5 embryos. **d–e** Analysis of differentially expressed genes in E15.5 Wt and *Nolz1⁻/⁻* mutant striatum by in situ hybridization validating the transcriptomic analysis. Striatonigral markers are downregulated (**d**) and several striatopallidal-specific markers are upregulated in *Nolz1⁻/⁻* mutant striatum (**e**). **f** A selection of striatopallidal, striatonigral and generic projection neurons markers that are differentially expressed between Wt and *Nolz1⁻/⁻* mutant striatum as identified by RNA sequencing. Graph represents fold change gene expression values in *Nolz1⁻/⁻* mutant striatum relative to Wt ($n = 3$ biologically independent samples). Expression values are presented as mean ± standard error log2 transformed values. **g** Expression validation by in situ hybridization of identified axon guidance molecules in E15.5 Wt and *Nolz1⁻/⁻* mutant striatum. Data are representative of three independent experiments (**d**, **e**, **g**). Scale bar (**d**, **e**, **g**): 200 μm. str (striatum), cx (cortex). Source data are provided as a source data file.

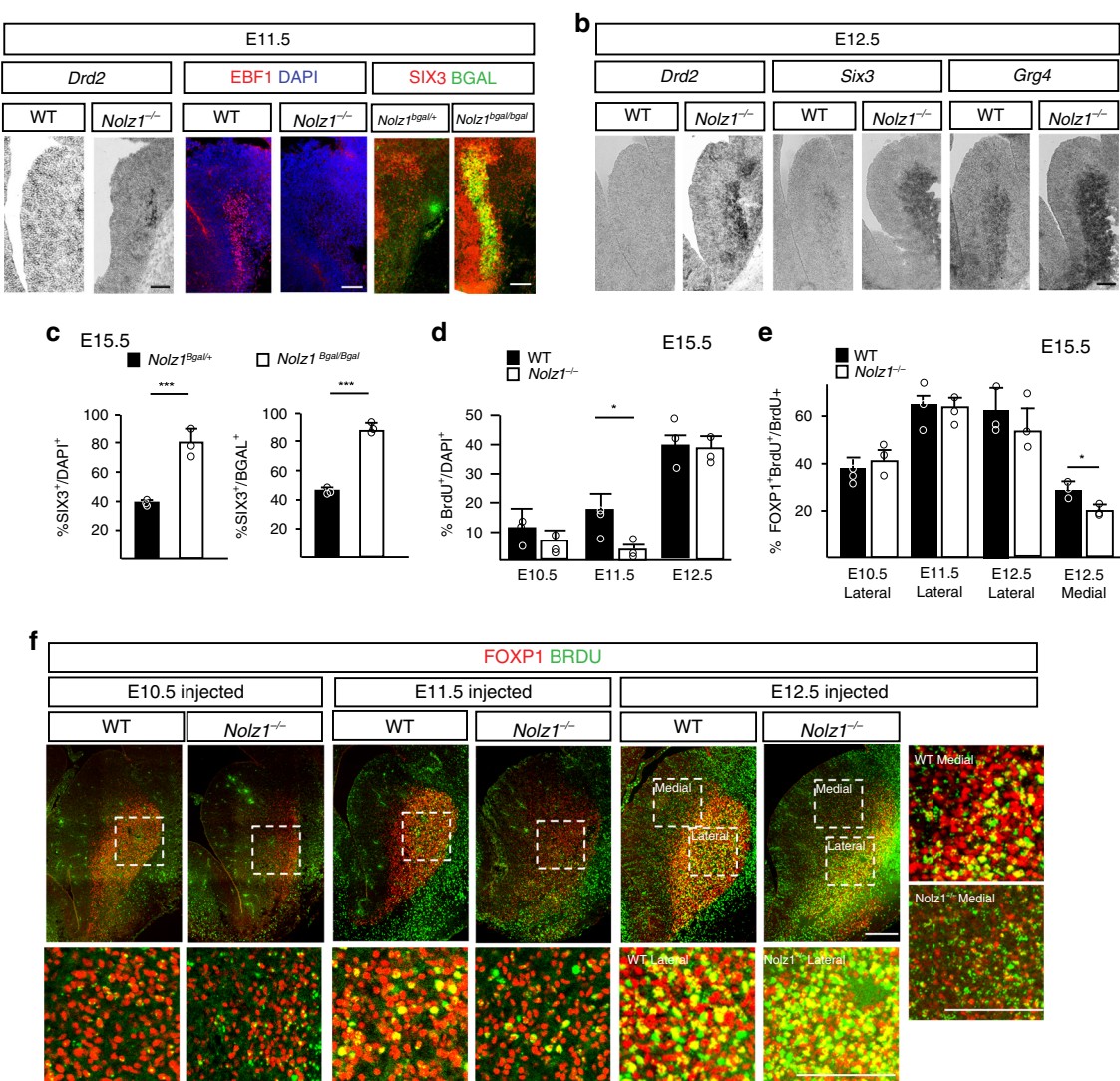

**Fig. 4 Temporal generation of striatal projection neurons is altered in *Nolz1*$^{-/-}$ mutant embryos. a** Expression of striatopallidal-specific markers *Drd2* and SIX3 at the expense of striatonigral marker EBF1 in E11.5 *Nolz1*$^{-/-}$ mutant embryos. Coronal sections of the striatum showing *Drd2* expression by in situ hybridization and EBF1$^+$DAPI$^+$ and SIX3$^+$bGAL$^+$ double labelled cells by immunofluorescence. Note the selective upregulation of SIX3 in cells lacking NOLZ1 expression indicated by bGAL in *Nolz1*$^{bgal/bgal}$ mutants. **b** Characterization of the expression profile of striatopallidal-specific markers in E12.5 Wt and *Nolz1*$^{-/-}$ mutant embryos by in situ hybridization. **c** Graphs showing the percentage of SIX3$^+$ cells versus DAPI (left graph) and bGAL labelled (right graph) cells in E15.5 *Nolz1*$^{bgal/+}$ and *Nolz1*$^{bgal/bgal}$ mutant striatum. Mean values ± standard deviation; $n = 3$ biologically independent samples; SIX3$^+$/ DAPI$^+$: Two-sided, unpaired *T*-test ***$p = 0.00273$, SIX3$^+$/bGAL$^+$: Two-sided, unpaired *T*-test ***$p = 6.79688 \times E^{-05}$. **d** Percentage of BrdU labelled cells in E15.5 Wt and *Nolz1*$^{-/-}$ mutant striatum after being injected with BrdU at different time points (E10.5, E11.5 or E12.5). Mean values ± standard deviation; $n = 3$ biologically independent experiments; Two-sided, unpaired *T*-test *$p = 0.0111$. **e** Graph showing percentage of FOXP1 expressing BrdU labelled cells in striatum of E15.5 Wt and *Nolz1*$^{-/-}$ mutant embryos injected with BrdU at E10.5, E11.5 or E12.5. Mean values ± standard deviation; $n = 3$ biologically independent experiments; Two-sided, unpaired *T*-test *$p = 0.02795$. **f** Immunofluorescence analysis of FOXP1 and BrdU expression in coronal sections of E15.5 Wt and *Nolz1*$^{-/-}$ mutant striatum injected with BrdU at different time points (E10.5, E11.5 or E12.5). Data are representative of three independent experiments (**a**, **b**, **f**). Scale bar in **a**–**f**, upper panel: 200 μm; Scale bar in **f** lower and right panels: 100 μm. Source data are provided as a source data file.

*Nolz1*$^{fl/fl}$ conditional mouse line with the midbrain (*En1Cre*), hypothalamic (*FoxD1Cre*) and telencephalic (*FoxG1-IRES-Cre*) specific *Cre* lines[33–35] (Fig. 5a). *Nolz1* was selectively ablated from the different regions using the respective *Cre* lines at E11.5 (Supplementary Fig. 5a). While no phenotypic alteration are observed in E18.5 control embryos (*Cre*$^+$, *Nolz1*$^{fl/fl}$ or *Cre*$^+$; *Nolz1*$^{fl/+}$) (Fig. 5b–b''), the selective ablation of *Nolz1* in the striatum of *FoxG1-IRES-Cre;Nolz1*$^{fl/fl}$ mutant embryos (Fig. 5c–c'') resulted in a DA axon defect highly similar to the phenotype observed in the constitutive *Nolz1*$^{-/-}$ mutant embryos. There was a strong reduction in striatal innervation by DA terminals (Fig. 5c, g) and a subset of DA axon bundles crossed the midline (Fig. 5c'),

although to a lesser extent compared with *Nolz1*$^{-/-}$ mutant embryos. In contrast, the deletion of *Nolz1* in either the hypothalamus (*FoxD1Cre;Nolz1*$^{fl/fl}$) (Fig. 5d–d'' and Supplementary Fig. 5g) or midbrain (*En1Cre;Nolz1*$^{fl/fl}$) (Fig. 5e–e'' and Supplementary Fig. 5g, h) did not affect the guidance of DA axons towards the striatum or striatal innervation (Fig. 5g), demonstrating the striatal-specific requirement of Nolz1 expression for establishing the DA circuitry. Furthermore, *FoxG1-IRES-Cre; Nolz1*$^{fl/fl}$ mutant embryos displayed similar phenotypic alterations as observed in *Nolz1*$^{-/-}$ mutant embryos, including the striatonigral to -pallidal fate change (Fig. 5h), altered expression of striatal axon guidance markers (Fig. 5i), the absence of

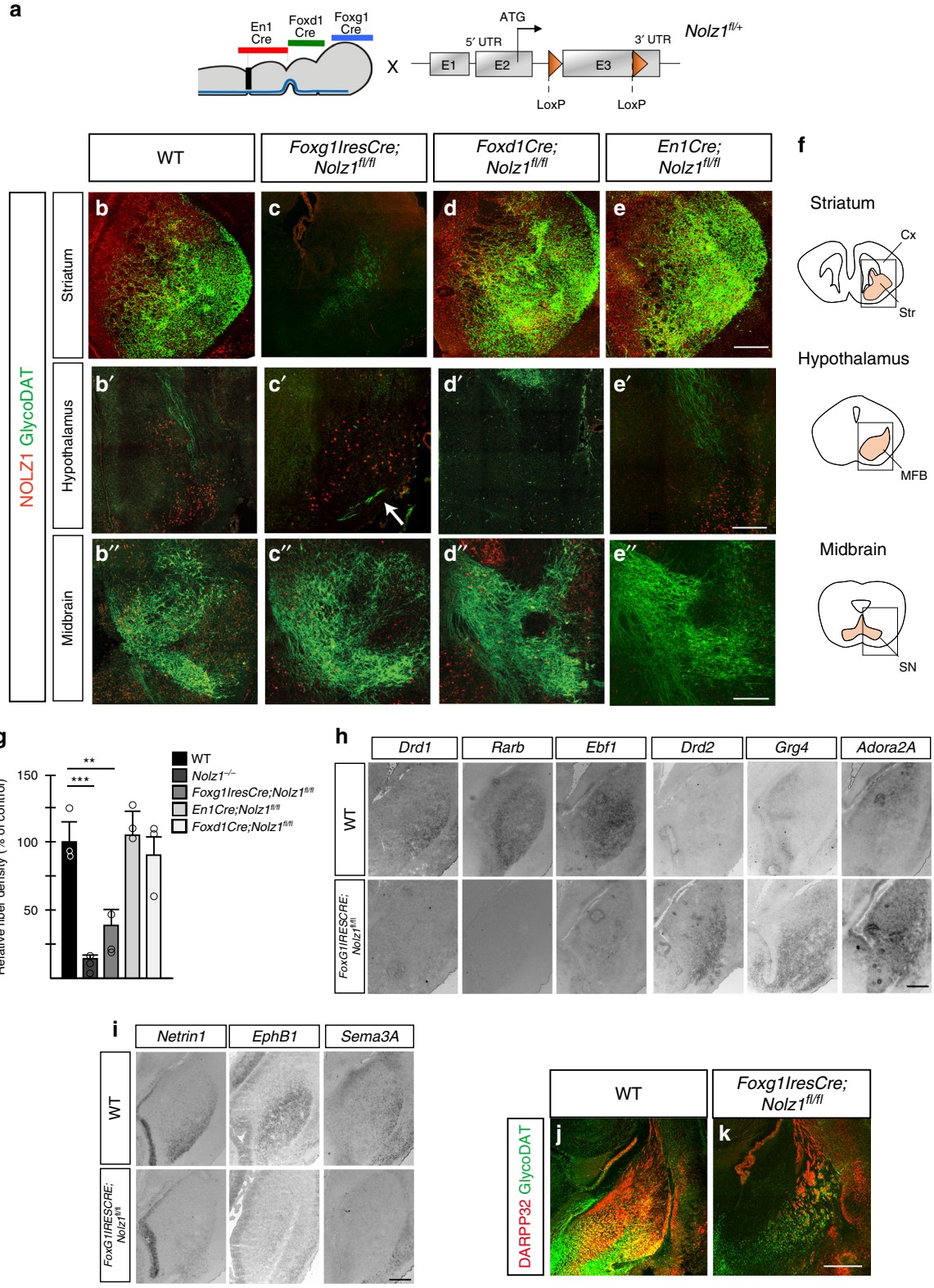

DARPP32 expression (Fig. 5j, k), lack of innervation of the GP (Supplementary Fig. 2a) and cCASP3 expression around the GP (Supplementary Fig. 2d). In contrast, there were no gene expression changes observed in the *EnCre;Nolz1fl/fl* mutant striatum (Supplementary Fig. 5i). Furthermore, the innervation of the prefrontal cortex was normally initiated in E18.5 *EnCre;Nolz1fl/fl* mutant embryos, while the fibre density of the nucleus accumbens was not changed in adults (Supplementary Fig. 5d–f).

Despite, the striatal selective ablation of *Nolz1*, born *Foxg1-IRES-Cre;Nolz1fl/fl* mutant pups died within 1 month after birth for unknown reasons.

**Nolz1 regulates establishment of axonal forebrain tracts.** The striatum is important for the establishment of several other forebrain axonal tracts, including thalamocortical and

**Fig. 5 DA axon guidance phenotype is mediated by lack of striatal *Nolz1* expression. a** Schematic representation of conditional *Nolz1* locus and *Cre* deleter lines used to ablate *Nolz1* in the following regions: *En1Cre* for midbrain, *FoxD1Cre* for hypothalamic and *FoxG1-IRES-Cre* line for telencephalic selective recombination. **b–e** Immunostaining for NOLZ1 and GlycoDAT on coronal sections of E18.5 mouse striatum, hypothalamus and midbrain in the following mouse lines: Control (**b–b''**), *FoxG1-IRES-Cre;Nolz1 fl/fl* (**c–c''**), *Foxd1Cre;Nolz1 fl/fl* (**d–d''**) and *En1Cre;Nolz1 fl/fl* (**e–e''**) mutant embryos. Arrow in **c'** points to DA axon bundles crossing the midline. **f** Representative outline of coronal sections from E18.5 mouse striatum (top), hypothalamus (middle) and midbrain (bottom) indicating regions represented in **b–e**. Cx (cortex), Str (striatum), MFB (medial forebrain bundle), SN (substantia nigra). **g** Graph showing GlycoDAT fibre density in the striatum of E18.5 *Nolz1*$^{-/-}$, *FoxG1-IRES-Cre;Nolz1 fl/fl*, *FoxD1Cre;Nolz1 fl/fl* and *En1Cre;Nolz1 fl/fl* mutant embryos relative to Wt embryos ($n = 3$ biologically independent samples). Wt values are normalized to 100%. Mean values ± standard deviation. Wt versus *Nolz1*$^{-/-}$**:** Two-sided, unpaired $T$-test ***$p = 0.00098$. Wt versus *FoxG1-IRES-Cre;Nolz1 fl/fl***:** Two-sided, unpaired $T$-test **$p = 0.00605$. **h** Analysis of striatonigral and striatopallidal-specific markers in control and *FoxG1-IRES-Cre;Nolz1 fl/fl* mutant embryos by in situ hybridization. Coronal striatal sections of E15.5 embryos. **i** Downregulation of axon guidance markers in E15.5 in *FoxG1-IRES-Cre;Nolz1 fl/fl* mutant embryos as shown by in situ hybridization on coronal sections. **j–k** Reduction of DARPP32 expression and innervation by GlycoDAT labelled axons in the striatum of E18.5 *FoxG1-IRES-Cre Nolz1 fl/fl* mutant embryos visualized by immunohistochemistry on coronal sections. Data are representative of three independent experiments (**b–e**, **h–k**). Scale bar (**b–k**): 200 μm. Source data are provided as a source data file.

corticothalamic projections[36–39]. Therefore, we investigated whether striatal absence of Nolz1 causes misrouting of other axonal tracts. Both Neurofilament (NF) and L1 cell adhesion molecule (L1) labelled axonal tracts are disorganized and fasciculate abnormally in *Nolz1*$^{-/-}$ and *FoxG1-IRES-Cre;Nolz1*$^{fl/fl}$ striatum of E15.5 and E18.5 mutant embryos (Supplementary Fig. 6a–i). In addition, L1 labelling shows that thalamocortical axons are normally formed in *Nolz1*$^{-/-}$ mutant embryos (Supplementary Fig. 6p, q), but fail to extend into the striatum (Supplementary Fig. 6j, l). In addition, DiI injected in the thalamus further reveals that thalamic axonal extension project ventrally instead of projecting towards the striatum in *Nolz1*$^{-/-}$ mutant embryos (Supplementary Fig. 8d). Within the thalamic region we observe a strong increase in cCASP3 expression in L1 labelled thalamic axons that project towards the striatum in *Nolz1*$^{-/-}$ mutant embryos at E18.5 (Supplementary Fig. 6m–p), but not in other axonal populations (Supplementary Fig. 6o, p). *FoxG1-IRES-Cre;Nolz1*$^{fl/fl}$ mutant embryos, which have a similar axon guidance phenotype, only show a minor induction of cCASP3 in the thalamic region (Supplementary Fig. 6r, s). Overlap between cCASP3 and bGAL expression in *Nolz1*$^{-/-}$ mutant cells (Supplementary Fig. 6q) indicate a cell-autonomous role of Nolz1 in regulating apoptotic marker expression. The phenotypic resemblance between the constitutive and conditional striatal-specific *Nolz1* mutant mouse lines demonstrates striatal, non-cell-autonomous requirement of Nolz1 in orchestrating the attraction and guidance of DA and other axonal tracts through the striatum.

Defects in formation of striatal axonal extensions has been linked to abnormalities in forebrain axonal tract formation[36]. To investigate whether the striatal outgrowth is affected in *Nolz1*$^{-/-}$ mutant embryos we injected DiI in E15.5 wild-type and mutant striatum (Fig. 6a–d). DiI injected in the wild-type striatum was retrogradely transported to the midbrain in Wt embryos (Fig. 6a, b), but not in *Nolz1*$^{-/-}$ mutant embryos (Fig. 6c, d) revealing a defect in the establishment of striatonigral projections (Fig. 6c). Consistent with the absence of striatonigral projections, the axonal length in striatal explants from *Nolz1*$^{-/-}$ mutant embryos was significantly shorter compared with wild-type explants (Fig. 6e, f).

**Nolz1$^{-/-}$ mutant striatum secretes cues repulsive to DA axons.** Next we investigated whether either reduced striatal axonal outgrowth or altered guidance cue expression could underlie the defects in establishment of DA neuronal connectivity observed in *Nolz1*$^{-/-}$ mutant embryos. In the absence of Pcdh10 striatal axonal extensions failed to form[36], a phenotype similar to that observed in *Nolz1*$^{-/-}$ mutant embryos. The lack of striatal axonal

outgrowth in *Pcdh10*$^{-/-}$ mutant embryos caused defects in the formation thalamocortical projections as shown by L1 and NF labelling (Supplementary Fig. 7b, c)[36]. Furthermore, Darpp32 expressing striatonigral axons failed to outgrow towards the midbrain (Fig. 6g, h)[36]. Despite the reduced striatal axonal outgrowth, the innervation of the striatum by DA axon terminals was not affected (Fig. 6i, j) in E14.5 mutant embryos. Furthermore, in *Pcdh10*$^{-/-}$ mutant embryos both striatonigral and -pallidal projection neurons markers and genes involved in axon guidance are normally expressed (Supplementary Fig. 7a). Thus the lack of striatal outgrowth alone does not cause the aberrant DA axonal trajectory observed in *Nolz1*$^{-/-}$ mutant embryos.

The transcriptomic analysis revealed that a subset of genes involved in axon guidance were altered in *Nolz1*$^{-/-}$ mutant striatum, including *Netrin1*, *Sema3a*, *Vegfc*, *PlxnD1* and *Tgfa* (Fig. 3g, Supplementary Fig. 3a, Supplementary Table S2 and Supplementary data 1). Therefore, we investigated whether the altered composition of secreted factors from the striatum influences the guidance of DA axons towards the striatum. There was no difference in axonal length and direction of outgrowth between wild-type and *Nolz1*$^{-/-}$ mutant ventral midbrain explants when cultured alone (Supplementary Fig. 8a). DA axons labelled by B-TUBULIN and TH extended from wild-type and *Nolz1*$^{-/-}$ mutant midbrain explants were attracted by wild-type striatal explants[40] (Fig. 6k–m; P/D ratio > 1). However, when midbrain explants were cocultured with *Nolz1*$^{-/-}$ mutant striatal explants DA axons were strongly repelled[40] (Fig. 6k–m; P/D ratio < 1). The distance between the explants was equal under all conditions (Fig. 6n). Similarly, axonal extension emanating from the thalamus were repulsed by striatal explants obtained from *Nolz1*$^{-/-}$ mutant embryos (Supplementary Fig. 8b, c). In contrast to midbrain explants, *Nolz1*$^{-/-}$ mutant thalamic explants were also repulsed by wild-type striatal explants indicating a thalamic requirement of Nolz1 in mediating chemoattraction.

The explant cocultures suggest that secreted factors emanated by *Nolz1*$^{-/-}$ mutant striatum[5] exerts a repulsive effect on DA axons. In agreement, conditioned medium collected from cultured *Nolz1*$^{-/-}$ mutant striatal explants caused collapsed growth cones on nearly 70% of the axons derived from ventral midbrain explants (Fig. 7a, b)[41], showing that factors secreted from the mutant striatum repulse DA axon outgrowth. Conditioned medium from Wt striatal explants did not have any effect on the growth cone morphology (Fig. 7a, b; 24% vs. 20% in controls). Of the secreted factors differentially expressed between wild-type and *Nolz1*$^{-/-}$ mutant striatum (Fig. 3g) we found that SEMA3A caused more than 70% of the growth cones to collapse, while TGFA and NETRIN1 did not have any effect (Fig. 7a, b). To investigate further whether there is a difference in the potential to attract DA axons over longer distances between wild-

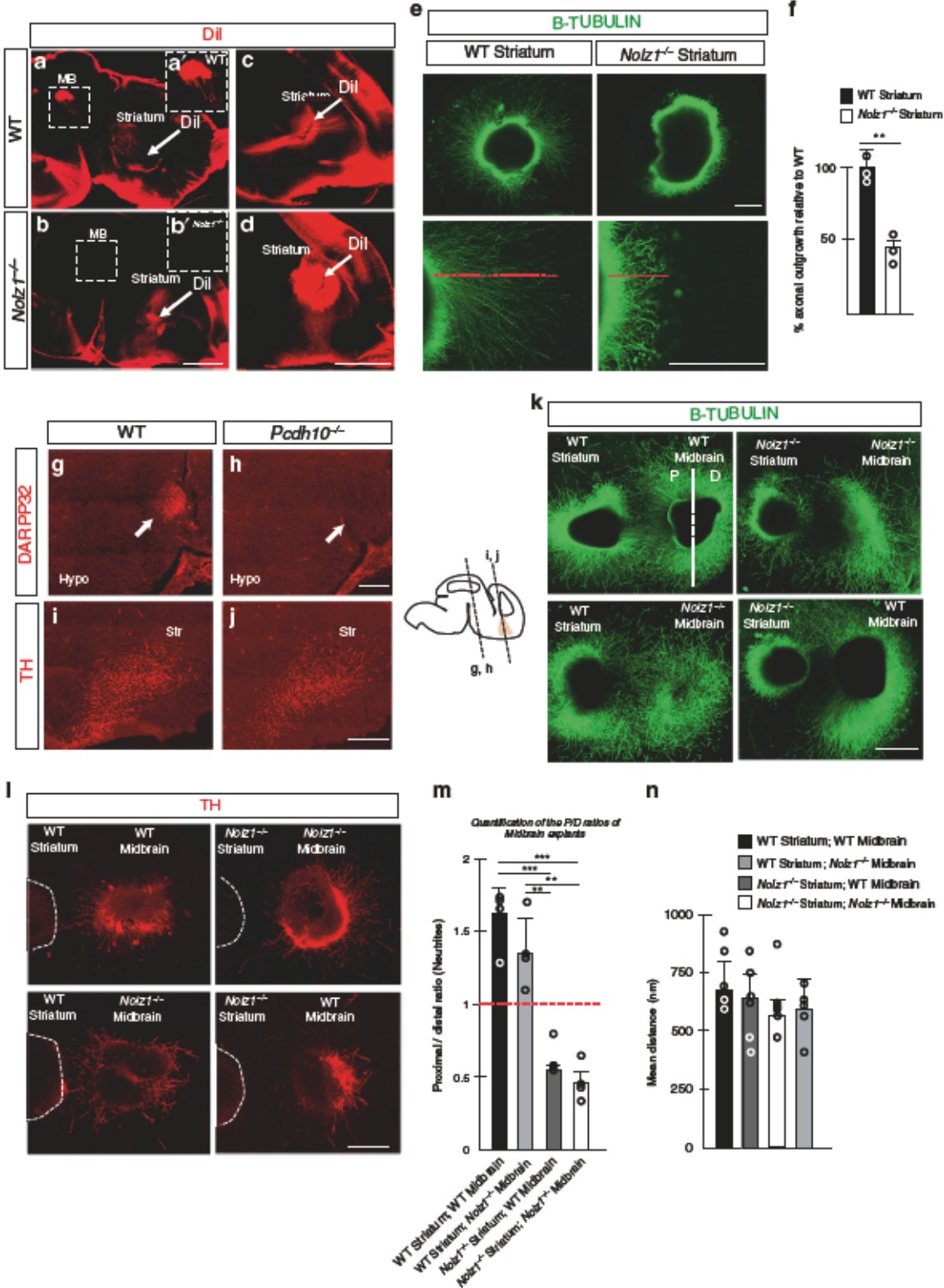

type and *Nolz1⁻/⁻* mutant striatum, we cultured wild-type ventral midbrain derived primary neurons and striatal explants in distinct compartments of a microfluidic device that are separated by microgrooves (Fig.7d). Wild-type striatal tissue attracted DA axons from the midbrain compartment. (Fig. 7e, j and Supplementary Fig. 8e). However, the presence of striatal explants derived from *Nolz1⁻/⁻* mutant completely abolished DA axon outgrowth (Fig. 7f, j and Supplementary 8e). *Tgfa* is one of the

candidate genes involved in the attraction of DA axons and the receptors EGFR and ERBB4, which bind and mediate TGFA signalling[42] are expressed in midbrain DA neurons (Fig. 7c). The addition of TGFA to the compartment containing *Nolz1⁻/⁻* mutant striatal explants completely restored the DA axonal outgrowth (Fig. 7h, j and Supplementary Fig. 8e), while it had no additional effect on the attraction of DA axons towards wild-type striatal explants (Fig. 7g). Similarly, TGFA could restore axon

**Fig. 6 Impaired striatal axonal outgrowth and repulsion of DA axons by *Nolz1*$^{-/-}$ mutant striatum. a–d** DiI injection in the striatum of E15.5 Wt and *Nolz1*$^{-/-}$ mutant embryos. Images display the failure of striatonigral neurons to project to the midbrain in *Nolz1*$^{-/-}$ mutant embryos (**a–b**). **c–d** reduced striatal axonal outgrowth in *Nolz1*$^{-/-}$ mutant mouse embryos. White arrows show the site of DiI injection. MB (midbrain). **e** Axon outgrowth from E13.5 Wt and *Nolz1*$^{-/-}$ mutant striatal explants in vitro. **f** Graph showing the percentage difference in axonal outgrowth from *Nolz1*$^{-/-}$ mutant striatal explants compared with Wt. Mean ± standard deviation; n = 3 biologically independent experiments; Two-sided, unpaired *T*-test **p = 0.00475. **g–h** Immunostaining of DARPP32 showing the lack of striatonigral projections in the hypothalamus of E14.5 *Pcdh10*$^{-/-}$ mutant embryos on coronal sections. **i–j** Immunostaining of TH on coronal sections of the striatum of E14.5 Wt and *Pcdh10*$^{-/-}$ mutant embryos. **k** Striatal and midbrain explants obtained from Wt and *Nolz1*$^{-/-}$ mutant embryos were cultured for 3 days and stained with b-TUBULIN (**k**) and TH (**l**). The proximal (P) part midbrain explant is facing the striatal explant, while the distal (D) compartment is furthest away from it. **m** Quantification of the neurite length in both the proximal and distal compartment of the explant culture assay as shown in (**k**). Graph shows P/D ratio. Mean values ± standard deviation; n = 4 biologically independent experiments; Two-sided, unpaired *T*-test: Wt Striatum + Wt Midbrain versus *Nolz1*$^{-/-}$ Striatum + Wt Midbrain ***p = 0.000135; Wt Striatum + Wt Midbrain versus *Nolz1*$^{-/-}$ Striatum + *Nolz1*$^{-/-}$ Midbrain ***p = 0.00013; Wt Striatum + *Nolz1*$^{-/-}$ Midbrain versus *Nolz1*$^{-/-}$ Striatum + Wt Midbrain **p = 0.00139; Wt Striatum + *Nolz1*$^{-/-}$ Midbrain versus *Nolz1*$^{-/-}$ Striatum + *Nolz1*$^{-/-}$ Midbrain **p = 0.00103. **n** Quantification of mean distance between midbrain and striatum explants across all conditions. Mean values ± standard deviation; n = 5 biologically independent experiments. Data are representative of three (**a–e**, **g–j**) or four (**k**) independent experiments. Scale bar 1000 μm (**a–b**), 500 μm (**c–e**, **k**, **l**), 200 μm (**g–j**). Hypo (hypothalamus) and Str (Striatum). Source data are provided as a source data file.

outgrowth from mES cell derived DA neurons (Supplementary Fig. 8f). However, the axonal attraction mediated by wild-type striatal tissue was totally abolished in the presence of the ERBB inhibitor Afatinib[43] (Fig. 7i, j and Supplementary Fig. 8e). These data demonstrate that TGFA signalling activation is sufficient to attract DA axonal projections.

## Discussion

The correct establishment and maintenance of DA axonal projections is important for normal brain function and defects in these processes underlie neurological and neurodegenerative diseases including Schizophrenia and Parkinson's disease[44,45]. Here we showed how striatal patterning defects in *Nolz1*$^{-/-}$ mutant embryos have an impact on the establishment of DA axonal projections, which led us to propose the following model as depicted in Fig. 8. (1) In both constitutive and striatal-specific conditional *Nolz1* mutant embryos a subset of DA axons crosses the ventral midline in the hypothalamus. The remaining axons maintain their trajectory towards rostral brain areas, but terminate in front of the striatum. (2) Furthermore, in the absence of striatal Nolz1 expression striatopallidal projections neurons are produced at the expense of striatonigral neurons. This lineage switch resulted in reduced striatal outgrowth (3) and altered composition of growth factors secreted from the striatum (4). The guidance of DA axons towards target areas does not depend on the elongation of striatal axons (3). Instead, the impaired striatal projection neuron specification and striatonigral–pallidal subtype conversion create a repulsive environment for DA growth cones and other forebrain tracts (4). Thus, we demonstrated that defects in striatal projection neuron specification have a direct effect on DA axonal tract formation and striatal innervation.

We showed that Nolz1, which is expressed in all striatal projection neurons, is selectively required for the specification of striatonigral projection neurons. In *Nolz1*$^{-/-}$ mutant embryos the expression of several striatonigral specific genes are either reduced or absent, which is accompanied by the ectopic induction of striatopallidal-specific genes. Nolz1 acts upstream of several other transcription factors that have been previously implicated in regulating the striatonigral fate. However, the striatonigral selective transcriptional regulators Isl1, Ebf1, Foxo1 and Rarb[13–17,46], regulate only a subset of striatonigral specific genes resulting in a much milder striatal phenotype. Although most of the striatopallidal markers are normally induced in the *Nolz1*$^{-/-}$ mutant striatum, the pallidal fate is not fully executed as shown by the reduction of *Darpp32*, *Arpp21* and *Gpr6* expression and the reduced innervation of the GP by striatopallidal neurons. Thus, while mainly the generation of striatonigral neurons is affected in

the absence of striatal Nolz1 expression, Nolz1 might also be required to regulate certain aspects of the striatopallidal fate. We found that in the absence of striatal Nolz1 expression, striato-pallidal projection neurons are generated at the expense of striatonigral neurons. As a transcriptional repressor[19,47–49] Nolz1 might promote the specification of nigral neurons by directly repressing striatopallidal-specific genes. A partial nigral to pallidal lineage switch is also observed in *Isl1*$^{-/-}$ mutant embryos[15] and Nolz1 might interact with Isl1 to prevent that pallidal transcriptional programmes are unsilenced in nigral neurons. The downregulation of striatonigral specific genes in the Nolz1 mutant striatum might be a consequence of the upregulated expression of pallidal genes in nigral neurons. Candidates for mediating the repression of nigral-specific gene programmes are *Six3* and *Tle4*[50,51]. Both genes are transcriptional repressors that have been previously implicated in pallidal fate specification[52] and are ectopically expressed in *Nolz1*$^{-/-}$ mutant striatum. In contrast to Nolz1 and Isl1, which are involved in the repression of pallidal genes, histone methyltransferase G9a is required to prevent the ectopic induction of striatonigral specific genes[53]. Thus, the specification of these two distinct projection neuron subtypes is at least to some extent dependent on the repressing activities of nigral and pallidal lineage determining factors. Our data also revealed that Nolz1 is involved in the timing of striatal projection neuron production, which could influence the subtype identity as well. BrdU birthdating experiments showed a strong decrease in neuronal birth at E11.5 in *Nolz1*$^{-/-}$ mutant embryos, which coincides with the production of EBF1 positive striatal projection neurons. Since Nolz1 expression is restricted to post-mitotic neurons, it is not clear how Nolz1 regulates cell cycle exit in the ventricular zone. A possibility is that Nolz1 influence the temporal aspects of projection neuron production indirectly through the induction of other secreted factors.

DA axons are guided to their target areas by temporal and spatial exposure to repulsive and attractive cues. Explant studies have revealed that DA axons are initially attracted by the MFB, followed by the striatum at later stages[5,54]. The initial recruitment of DA axons towards the MFB is not affected in *Nolz1*$^{-/-}$ mutant embryos. However, the subsequent progression of DA axonal projections into a rostral direction towards the striatum is disrupted, resulting in midline crossing of a subset of DA axons and lack of striatal innervation. Although gene mutations in *Netrin*, *Slit2*, *Robo1*, *Dcc* and *Nrp2*[55–59] also causes aberrant midline crossing, these guidance cues are broadly expressed and their tissue selective requirement has remained unclear. Our results show for the first time that midline crossing of DA axons in the hypothalamus is mediated by phenotypic alterations in the

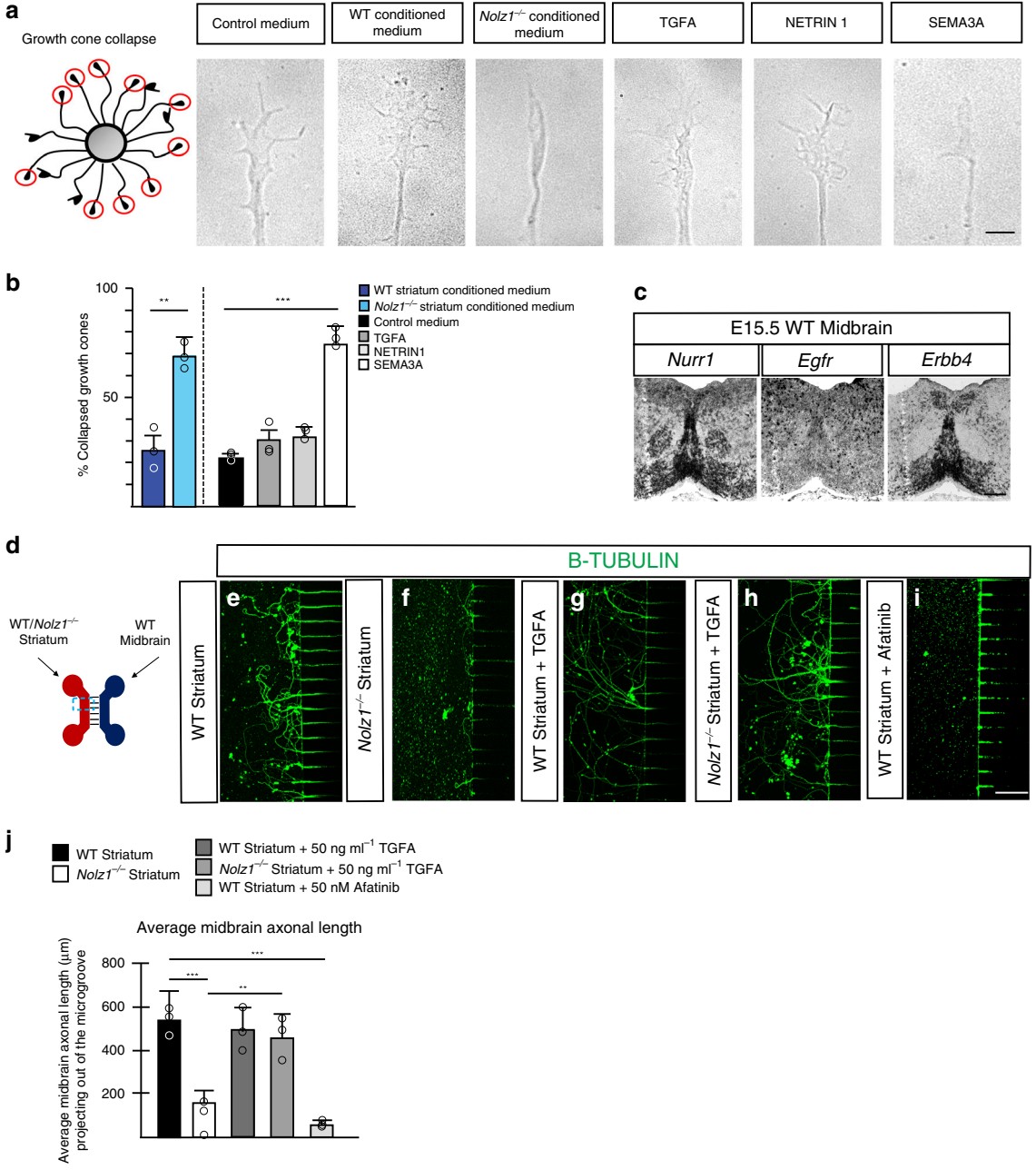

**Fig. 7 *Nolz1⁻/⁻* mutant striatum repulses DA axons. a** Schematic representation of scoring system. Red circles indicate collapsed growth cones. Representative images of growth cones responses to control medium, E13.5 Wt and *Nolz1⁻/⁻* mutant-derived striatal conditioned medium, 50 ng ml⁻¹ TGFA, 300 ng ml⁻¹ NETRIN1 and 300 ng ml⁻¹ SEMA3A. **b** Graph showing the percentage of collapsed growth cones exposed to different conditions (*n* = 3 independent experiments). Mean values ± standard deviation; Two-sided, unpaired *T*-test: Wt striatum versus *Nolz1⁻/⁻* striatum **\*\*$p$ = 0.00237; Control medium versus SEMA3A medium \*\*\*$p$ = 1.43231xE⁻⁰⁵. **c** In situ hybridization showing expression of *Egfr*, *Erbb4* and *Nurr1* in E15.5 Wt ventral midbrain (coronal sections). **d–i** Microfluidic assay to assess attractive and repulsive effects of Wt and *Nolz1⁻/⁻* mutant striatal tissue and TGFA signalling on DA axons. Axons were labelled by b-TUBULIN. **d** Design of microfluidic platform. Primary DA neuronal cultures obtained from E13.5 Wt embryos were seeded in the cellular compartment and E13.5 Wt or *Nolz1⁻/⁻* mutant striatal explants were cultured in the opposing compartment. Primary DA neurons were cultured in the presence of either Wt striatal explants (**e**), *Nolz1⁻/⁻* mutant striatal explants (**f**), Wt striatal explants with 50 ng ml⁻¹ TGFA (**g**), *Nolz1⁻/⁻* mutant striatal explants with 50 ng ml⁻¹ TGFA (**h**) or Wt striatal explants with 50 nM Afatinib (**i**). **j** Graph showing the average axonal length of midbrain neurons projecting out of the microgroove under different conditions Mean values ± standard deviation; *n* = 6 biologically independent experiments; Two-sided, unpaired *T*-test: Wt Striatum versus *Nolz1⁻/⁻* Striatum \*\*\*$p$ = 3.59072xE⁻⁰⁶; *Nolz1⁻/⁻* Striatum versus *Nolz1⁻/⁻* Striatum + TGFA **\*\*$p$ = 0.0017; Wt Striatum versus Wt Striatum + Afatinib \*\*\*$p$ = 1.74752xE⁻⁰⁶. Data are representative of three (**a**, **c**) or six (**d–i**) independent experiments. Scale bar (**a–i**): 200 μm. Source data are provided as a source data file.

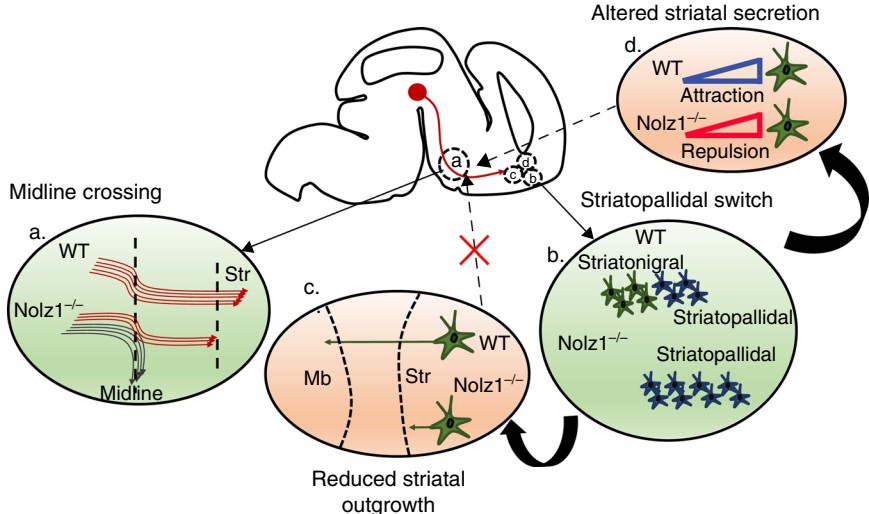

**Fig. 8 Model depicting the *Nolz1*$^{-/-}$ mutant phenotype. a**. A subset of DA axons crosses the ventral midline of the hypothalamus, while remaining axon bundles fail to innervate the striatum in *Nolz1*$^{-/-}$ mutant embryos. **b** Striatopallidal projection neurons are born at the expense of striatonigral neurons in *Nolz1*$^{-/-}$ mutant striatum. The striatonigral–pallidal lineage switch causes (**c**) reduced striatal axonal outgrowth and (**d**) altered composition of growth factors secreted from the striatum in the mutant embryos generating a repulsive environment for DA axons. The loss of attraction (**d**) causes DA axon guidance phenotype as described in (**a**), while reduced axonal outgrowth alone has no impact on DA axons.

striatum. Firstly, in *Nolz1*$^{-/-}$ mutant embryos gene expression changes were only observed in the striatum, but not in other brain regions DA axons are guided through. Secondly, the region selective ablation of *Nolz1* further confirmed that the misguidance of DA axons is primarily a consequence of the absence of striatal Nolz1 expression. Although *FoxG1-IRES-Cre;Nolz1*$^{fl/fl}$ conditional mutant embryos display similar DA axon guidance defects as observed in *Nolz1*$^{-/-}$ mutant embryos, the phenotype in the conditional mutants is milder. In contrast to DA axons, the importance of the striatum in the formation of forebrain axonal tracts has been demonstrated before. For example, the shortening of striatal axonal extensions in *Celsr3*$^{-/-}$ and *Pcdh10*$^{-/-}$ mutant embryos[36,60] impaired the establishment of thalamocortical axonal projections. However, whether and how the specification of striatal projection neurons influences the guidance and innervation of DA axons has not been investigated in much detail. While striatal defects observed in *Nolz1*$^{-/-}$ mutant embryos totally abolished DA innervation, in other mutants with striatal patterning defects the striatum remains innervated by DA axons. For example, in *Gsh1/2* double mutant embryos[61] in which striatal development is severely affected, DA terminals still enter the striatum. Also, the striatum still receives DA axonal innervation in *Ebf1*$^{-/-}$ and *Rarb*$^{-/-}$ mutant embryos in which the specification of striatonigral neurons is impaired[16,62,63]. In addition, while graded Netrin and Sema7a signalling in the striatum are required for the topographic guidance of DA axons, it is not necessary for DA innervation[57,64]. Thus, the loss of Nolz1 expression results in a striatal phenotype that uniquely affects the guidance and innervation of DA axons. The striatonigral to -pallidal lineage switch observed in *Nolz1*$^{-/-}$ mutant embryos demonstrates the influence of striatal projection neuron subtype specification on establishment of DA projections. Analysis of *Pcdh10*$^{-/-}$ mutant embryos revealed that striatal axonal outgrowth is not required for directing DA projections towards forebrain target areas. Therefore, it is more likely that the altered composition of secreted factors emanated by *Nolz1*$^{-/-}$ mutant striatum causes aberrant DA axonal tract formation and loss of striatal innervation.

The striatum normally produces a permissive environment for axonal tracts that are guided through the forebrain[5,54,65].

However, extrinsic factors secreted from the *Nolz1*$^{-/-}$ mutant striatum impose a repulsive effect on DA axons as shown by the growth cone collapse assay and explant culture. Whether the repulsive environment is caused by the loss of attractive or by the induction of repulsive guidance factors in *Nolz1*$^{-/-}$ mutant embryos will need to be determined. It is likely that a combination of factors secreted by the *Nolz1*$^{-/-}$ mutant striatum causes the observed DA axon guidance phenotype. It has been shown that axonal responsiveness to guidance factors is temporally regulated and highly context dependent[66]. Also the induction of cCASP3 in thalamic axons could alter axonal responsiveness to extrinsic factors resulting in guidance defects[67]. We identified TGFA as one of the secreted signalling molecules that is downregulated in the striatum of *Nolz1*$^{-/-}$ mutant embryos. In *Tgfa*$^{-/-}$ mutant embryos there is a slight reduction of striatal innervation[68]. Interestingly, we showed that addition of TGFA could rescue the outgrowth of DA axons towards the *Nolz1*$^{-/-}$ mutant striatum. The TGFA receptors EGFR1 and ERBB4 are expressed by DA neurons and blocking the receptor signalling by Afatinib[43] totally abolished the axonal outgrowth. So far, TGFA signalling has been shown to regulate axonal growth indirectly through its function in astrocytes[69]. Our finding suggests a more direct role of this signalling pathway in regulating axonal growth and shows that TGFA is sufficient to attract DA axons towards the striatum. Detailed understanding of mechanisms involved in the establishment of DA axonal projections will facilitate the development of novel strategies to improve graft outcome in cell replacement studies. Our study has identified several signalling molecules altered in *Nolz1* mutant striatum that could be further exploited for its involvement in DA axon guidance. For example, addition of signalling molecules including TGFA could promote the outgrowth of DA axons from intranigral grafts and accelerate target innervation.

## Methods

**Mouse lines**. All mice were kept in standard conditions with food and water ad libitum and maintained on the C57BL/6J genetic background. The *Nolz1*$^{+/-}$ (*Nolz1*$^{bgal/+}$) heterozygous mutant mouse line was generated by Brooks (manuscript under preparation) in which the coding region of *Nolz1* was replaced by the *beta-Galactosidase* gene (Supplementary Fig. 1a). *Nolz1*$^{fl/+}$ heterozygous mutant mice were generated by Genoway by homologous recombination in mouse

embryonic stem (ES) cells. LoxP sites are flanking the coding region of exon 3 leaving the 3′UTR intact (Fig. 5a). The Neo cassette was removed by crossing Nolz1$^{fl/+}$ heterozygous mice with a Flp- recombinase expressing mouse line. The Cre-mediated excision enables the deletion of the loxP-flanked region, resulting in a Nolz1$^{-/-}$ knockout allele. The Cre deleter lines FoxD1Cre$^{/+}$ (012463) and En1Cre$^{/+}$ (007916) were obtained from The Jackson Laboratory. The FoxG1-IRES-Cre mouse line[33] was obtained from Dr. Miyoshi (University of Tsukuba, Japan). To obtain each individual conditional knockout line described in this study, Nolz1$^{fl/fl}$ animals were crossed with either FoxG1-IRES-Cre$^{/+}$;Nolz1$^{fl/+}$, FoxD1Cre$^{/+}$; Nolz1$^{fl/+}$, or En1Cre$^{/+}$;Nolz1$^{fl/+}$ animals to obtain FoxG1-IRES-Cre$^{/+}$;Nolz1$^{fl/fl}$, FoxD1Cre$^{/+}$;Nolz1$^{fl/fl}$, or En1Cre$^{/+}$;Nolz1$^{fl/fl}$ mutant offspring, respectively. The following primers were used for genotyping: forward Nolz1 bGal (5′-GTTGCAGTGACGGCAGATACACTTGCTGA-3′), reverse Nolz1 bGal (5′-GCCACTGGTGTGGGCCATAATTCAATTCGC-3′), forward Annexin-1 (5′-AGATGAAATTGGGTGCAAATTCTAAGGGG-3′), reverse Annexin-1 (5′-TGTAAATATACTAGCTTCTGAGGAAGGCGACTTTG-3′), forward Cre (5′-ATTGCTGTCACTTGGTCGTGGC-3′), reverse Cre (5′-GGAAAATGCTTC TGTCCGTTTGC-3′). Forward and reverse Annexin-1 primers were designed and validated by GenOway for the specific detection of the conditional knockout allele in Nolz1$^{fl/fl}$ conditional knockout mouse lines. The Pcdh10$^{+/-}$ heterozygous mouse line[36] was bred by Dr. Ferri (University of Iowa, USA) and used to obtain Pcdh10$^{-/-}$ mutant embryos. All animals were kept on a 12 h day/night cycle, at 50 ± 10% humidity and at 21 ± 1 °C. All animal procedures followed the guidelines and legislation as regulated under the Animals Scientific Procedures Act 1986 (ASPA) and were approved by the Animal Welfare and Ethical Review Body (AWERB) from the University of Leicester.

**Immunohistochemistry**. Embryonic tissue was fixed in 4% PFA on ice for either 1 h (E11.5 heads), 1.5 h (E13.5 heads and E15.5 brains) or 3 h (E18.5 dissected brains). Brains were embedded in OCT (VWR) and cryosectioned at a thickness of 12 µm using a Leica cryostat CM3050S. Sections were mounted on Superfrost plus glass slides (Thermo Scientific, USA) and air dried for 30 min and washed 3 × 5 min in PBS/0.1% triton-x. Adult brains were fixed in 4% PFA overnight at 4 °C and sections were cut on a freezing microtome (Leica) at a thickness of 30 µm. Immunohistochemistry was performed as described[18]. Nuclei were visualized by DAPI (Sigma). Stained cells were analyzed using a confocal microscope (LSM 510 Meta, Zeiss or LSM 880, Zeiss). The antibodies and concentrations are shown in the supplementary resource table.

**BrdU administration and tissue processing**. Time-mated pregnant females were injected subcutaneously with 5-bromo-2′-deoxyuridine (BrdU) 100 mg kg$^{-1}$ (Sigma Aldrich, USA) at E10.5, E11.5 or E12.5. Embryos were isolated at E15.5 and dissected brain were fixed for 1.5 h in 4% PFA on ice and further processed as for immunohistochemistry. For BrdU staining, sections were incubated with 0.1 M citric acid for 20 min at 100 °C, adjusted to room temperature and rinsed three times in PBS before applying the blocking solution containing 0.25% Triton X-100 (Sigma), 5% donkey serum in PBS (Jackson ImmunoResearch). Sections were incubated with rat monoclonal anti-BrdU antibody (1:500, Abcam) diluted in blocking solution at overnight at 4 °C. The next day, sections were washed three times in PBS before being incubated for 1 h with a goat anti-rat secondary antibody (1:500, Abcam).

**In situ hybridization**. Tissue was fixed and processed as described for the immunohistochemistry. Probes were selected from gene lists obtained by comparing gene expression profiles of E18.5 Wt and Nolz1$^{-/-}$ mutant striatum. Primers used for generating in situ probes are shown in the supplementary Table 3. The in situ probes were produced by in vitro transcription as using cDNA from E18.5 mouse embryos as a template. The in situ hybridization was performed following standard procedures. Sections were analyzed and photographed using Axiovert 200 microscope (Zeiss) equipped with a digital camera using 10x magnification.

**In vitro explant culture in collagen matrices**. Ventral midbrain, striatal and thalamic explants were obtained from E13.5 embryos as described[40]. Striatal explants were embedded in a collagen matrix either with an explant derived from the ventral midbrain or thalamus as previously described[40] with some minor modifications. Special care was taken to keep a distance of 300–500 µm (1 explant size) between explants during the embedding process. Explants were cultured in explant medium containing Neurobasal A medium supplemented with B27, 1 M Hepes, L-glutamine, Pen-Strep and Beta-mercaptoethanol for 3 days. To determine the attractive/repulsive effects on midbrain and thamamic explants, the length of b-Tubulin labelled axons was measured with Zeiss software 2012. For quantification, the midbrain and thalamic explants were divided into two areas, proximal (facing towards the explant) and distal (facing away from explant). In cases where explants extended many axons, only the five largest were considered. The average axonal length for the proximal and distal compartments was calculated to obtain the P/D ratio. When the value was >1.0, the effect was considered attractive; values <1.0 were considered to be indicative of a repulsive effect[40]. The distance between two

explants was measured using the Measure function in Zeiss software 2012 to control for aberrant distance-mediated effects on axonal outgrowth.

**Axonal growth chambers in microfluidic devices**. Polydimethylsiloxane microfluidic devices were purchased from Xona Microfluidics (Standard Neuron Device Cat.No: SND900) and were used according to the manufacturer's protocol. Briefly, they were attached by pressure to a 10 cm dish precoated with poly-D-lysine (50 µg mL$^{-1}$). Wt ventral Midbrain tissue was dissected from E13.5 mouse embryos, kept in 5% FCS + DMEM solution and dissociated with TrypLE Express (Gibco) to obtain a single-cell suspension of DA neurons. Cells were then plated (50,000 cells µL$^{-1}$) in the cell compartment of the microfluidic device. After 10 min, 150 µL of explant medium was added. The other side of the chamber was filled with 200 µL of explant medium with striatum explants from Wt (positive control) or Nolz1$^{-/-}$ mutant embryos. Nolz1$^{-/-}$ mutant striatal tissue was treated with 50 ng ml$^{-1}$ TGFa, while 50 nM Afatinib was added to wild-type striatal explants. Afatinib was dissolved in DMSO at a stock concentration of 100 mM and diluted to 50 nM using serial dilutions. DMSO was added to wild-type striatal explants as a control at a similar concentration. Devices were maintained at 37 °C in humidified 5% CO$_2$/95% air for 9 days and medium was changed regularly every 2–3 days. Cells were fixed by adding 200 µl of pre-warmed 4% PFA with 10% sucrose and stored at 4 °C for immunohistochemistry. Importantly, while cells were seeded at the same initial density, each chamber contained a variable number of axons after 9 days. Axonal length in the devices was measured using Zeiss ZEN black 2012 software by drawing a segmented line along each individual axon projecting out of the microgroove in the distal compartment.

**DiI injection**. Tracing of neural projection with DiI (1,1-dioctadecyl-3,3,3,3-tetramethyl-indocarbocyanine perchlorate; Molecular Probes) was done as described[36]. For labelling axonal striatal projections, we microinjected a DiL solution (100 µg ml$^{-1}$ in N,N-dimethylformamide) using microcapillare pressure into the striatum on fixed E15.5 brains. Brain injected were then incubated in 4% PFA in rotation during 4 weeks at RT. The brains were then sectioned by vibratome (Leica VT1200S) (150-µm section) and analyse directly by confocal microscope.

**Striatal conditioned medium**. The striatum was dissected from E13.5 mouse embryos in 5% FCS + DMEM solution and dissociated with TrypLE Express (Gibco) to obtain single cells DA neurons. Cells were plated (1 striatum/well) in 24 wells plate coated with poly-D-lysine (50 µg mL$^{-1}$). One millilitre of explant medium previously describe was added. Striatal conditioned medium was obtained after 7 days of striatal primary culture from striatum Wt or Nolz1$^{-/-}$ mutant. The media were collected and immediately placed on dry ice and then stored at −80 °C.

**Growth cone collapse assay**. Wells were coated overnight with poly-D-lysine (50 µg mL$^{-1}$, Sigma) at 4 °C, subsequently washed with sterile PBS three times before being coated with laminin (1:10 in PBS, Invitrogen, UK) and incubated for 2 h at 37 °C. In the meantime, ventral midbrain tissue was dissected from E13.5 WT mouse embryos, placed in ice-cold, Dulbecco's modified Eagle's medium (DMEM; Gibco-BRL), sectioned into small pieces and then placed into the centre of each laminin-coated well. Before seeding, wells were washed two times with sterile PBS and then supplemented with explant medium as previously described[40]. Explants were then cultured at 37 °C for 72 h to the laminin-coated wells. For control experiments, the explant medium was removed and replaced with fresh medium. Either conditioned medium derived from Wt and Nolz1$^{-/-}$ mutant striatal tissue or recombinant proteins (R&D) of the axon guidance cues NETRIN1, SEMA3A and TGFa were added to the cell medium at a concentration of 300 ng mL$^{-1}$, 300 ng mL$^{-1}$ and 50 ng mL$^{-1}$, respectively. Following addition of appropriate medium or recombinant protein, plates were incubated at 37 °C for 30 min. After the fixation (1 h in pre-warmed 4% PFA with 10% sucrose), explants were washed with PBS and stored at 4 °C. For quantification, growth cones were scored as either collapsed or uncollapsed using an inverted phase contrast microscope with a ×40 objective. According to previous published methods[41], tips of axons were classified as uncollapsed if prominently spread growth cones with flattened lamellipodia and/ or two or more filopodia were identified. Bullet-shaped neurite tips with less than two filopodia and/or no lamellipodium present were scored as collapsed growth cones.

**Image acquisition and quantification**. Images were acquired on an Axiovert 200 M inverted microscope, Zeiss LSM 880 or Zeiss LSM 510 confocal microscopes and captured at ×10, ×20 or ×25 objective depending on staining procedure, sample size and area of interest. Images were processed using Adobe Photoshop CC2018 and Adobe Illustrator CC2018 software. For quantification, high-definition tilescan confocal images were taken at each level of the striatum in three different animals per condition using the lateral ventricle as an anatomical landmark. Images were opened with ImageJ and converted to 8-bit. Positively labelled cells were identified using built-in threshold algorithms to best determine the intensity of the background against the positive staining. After applying the threshold settings, a specified region of interest was traced along the contours of the striatum and positively

labelled cells were quantified semi-automatically in ImageJ. Similarly, to quantify the total area of each striatum, the striatum was manually delineated in each single-plane section using the line selection tool and the total area was automatically computed across all rostro-caudal sections of the striatum. Results were converted from pixels$^2$ to mm$^2$ by spatially recalibrating each image. Microsoft Excel was used for the organization and statistical analysis of the data.

**Cell counting and fibre density analysis**. To determine the percentage of Aldh1a1 expressing TH neurons in the midbrain the number of ALDH1A1 and TH positive cells was counted manually in every 4th section of each E18.5 embryo in at least three embryos of each genotype. Statistical significance was calculated by the unpaired student's $t$ test and data is presented as mean ± sd. Striatal fibre density of GlycoDAT positively labelled axonal projections was measured by densitometry using ImageJ software. Striatal sections were fluorescently labelled with GlycoDAT and images were taken using confocal microscopy. Fibre density of whole striatal area was measured and the measured values were corrected for non-specific background staining by subtracting values obtained from the cortex. Of each embryo every 8th section was analyzed of at least three different embryos.

**iDISCO**. Dissected brains from E18.5 WT and Nolz1$^{-/-}$ mutant embryos were processed as previously described[22]. Briefly, whole brains were dehydrated in methanol in PBS and bleached overnight at 4 °C. Tissue was then gradually rehydrated in PBS by removing. Detergent washing was then performed in PBS with 0.2% Triton X-100 (2 × 1 hr). Tissue was incubated overnight at 37 °C in PBS with 0.2% Triton X-100 and 0.3 M glycine, followed by blocking in PBS with 0.2% Triton X-100 and 6% normal donkey serum for 2 days. Following blocking, the tissue was washed for 1 hr twice in PBS with 0.2% Tween-20 and 10 μg mL$^{-1}$ heparin (PTwH). Brains were incubated with a primary rabbit-anti-TH (1:400) antibody diluted in PTwH/5%DMSO/3% Donkey serum at 37 °C for 4 days. Excess primary antibody was washed for 1 day in PTwH with periodic solution changes. Secondary antibody donkey anti rabbit Alexa 647 (1/250) was applied in PTwH/3% Donkey serum at 37 °C for 4 days. After incubation with the secondary antibody, samples were washed in PTwH for 5 days with periodic solution changes. After washing away excess secondary antibodies, optical clearing of iDISCO samples was performed as described in ref. [22]. Tissue was gradually dehydrated in resistant glassware with tetrahydrofuran in water. Remaining lipids were extracted with dichloromethane for 1 hr and dibenzyl ether (DBE) was used for refractive index matching. Samples were kept in a full vial of DBE.

**3D imaging**. E18.5 Wt and Nolz1$^{-/-}$ mutant brain tissue were imaged in horizontal orientation with an Ultramicroscope II (LaVision BioTec) lightsheet microscope equipped an MVX-10 Zoom Body (Olympus), MVPLAPO ×2 Objective lens (Olympus), Neo sCMOS camera (Andor) (2560 × 2160 pixels. Pixel size: 6.5 × 6.5 μm$^2$) and Imspector (version 5.0285.0) software (LaVision BioTec). Samples were scanned with double sided illumination, a sheet NA of 0.148348 (results in a 5-μm-thick sheet) and a step-size of 2.5 μm using the horizontal focusing light sheet scanning method with the optimal amount of steps and using the contrast blending algorithm. The effective magnification for all images was 4304× (zoombody × objective + dipping lens = 2× × 2.152×). Following laser/filter combination was used: Coherent OBIS 647-120 LX laser with a 676/29 emission filter.

**RNA extraction and quantitative real-time PCR**. Total RNA was extracted using a RNeasy micro kit (Qiagen) according to the manufacturers' protocols. Biological replicates were obtained from at least three independent experiments. The RNA quality was assessed using a Nanodrop 2000 spectrophotometer (Thermo Scientific) prior to amplification. cDNA was synthesized from total RNA via reverse transcription using the reverse transcriptase SuperScriptIII (Invitrogen) and Oli-godT as primer (Invitrogen). Real-time qPCR was performed on a QuantStudio 6 Flex system (Thermo Fisher) using Fast SYBR Green Mastermix (Thermo Fisher). Gene expression values were normalized against RPL19 and fold change was calculated using the $2^{-\Delta\Delta CT}$ method. The primer sequences used are shown in the supplementary resource table. Measurement were taken from distinct samples. Significance was determined by the two-tailed student's $t$ test.

**RNA sequencing and bioinformatic analysis**. Striatal tissue was dissected from both hemispheres of E18.5 Wt and Nolz1$^{-/-}$ mutant embryos, frozen on dry ice and stored at −80 °C. Each sample containing both hemispheres from the same embryos was further processed for RNA extraction using RNeasy Micro Kit (Qiagen Inc., UK). Three replicates consisting both hemispheres were used for each biological condition.

**Library preparation**. Total RNA was first examined using the Agilent RNA 6000 Nano kit. All samples passed QC with a RIN value > 8. To remove ribosomal RNA 2.5 μg total RNA was treated with the Illumina RiboZero Gold (Human/Mouse/Rat) kit, according to the manufacturer's protocol. From 100–200 ng of the ribo-depleted RNA sequencing libraries were prepared using the Illumina TruSeq Stranded mRNA kit, according to the manufacturer's protocol. Sequencing libraries were examined using the Agilent High Sensitivity DNA kit, and library concentrations

determined. Sequencing was carried out by the Earlham Institute, 150PE using the HiSeq4000. Sequenced libraries were verified using FastQC (version 0.11.5).

**RNA-seq data analysis**. Adapters were trimmed using Cutadapt (version 1.16). Reads were mapped to the genome in a transcriptome-aware manner using HISAT2 (version 2.1.0), using the mouse HISAT2 genome tran index (Ensembl, GRCm38). Features were counted using the featureCounts function from the Rsubread package (version 1.28.1), using the annotation in the relevant Ensembl gtf file (GRCm38 release 92).

**Bioinformatics**. The differential expression between WT and mutant was analyzed using the DESeq2 (version 1.18.1) package. Genes were identified as being differentially expressed at an adjusted $p$ value of <0.05 for the DESeq2 results and with a fold change of at least 1.5. Genes passing these thresholds, in either method, were used in subsequent analyses. Using these criteria there were 166 genes down-regulated and 139 genes upregulated in the Nolz1$^{-/-}$ mutant striatum. The heatmap was created using the heatmap.2 function within the gplots R package using significantly changing genes from the DESeq2 analysis with a fold change of at least 2. Clustering was done within heatmap.2 using complete-linkage.

**Reporting summary**. Further information on research design is available in the Nature Research Reporting Summary linked to this article.

## Data availability
The transcriptomic data have been deposited to ArrayExpress with the data set identifier E-MTAB-8240 (https://www.ebi.ac.uk/arrayexpress/experiments/E-MTAB-8240/). All data are included in this article and supplemental data files and are available from the corresponding author upon reasonable request. The source data underlying Figs. 2u, v, w; 3f; 4c, d, e; 5g; 6f, m, n; 7b, j and Supplementary Figs. 1d, e; 2c; 4b, c, g; 8c can be found in the source data file.

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

## Acknowledgements

We would like to thank Dr. Miyoshi for providing the *FoxG1-IRES-Cre* line and Dr. Hirano and for access to the *Pcdh10* mutant line. We would like to acknowledge the help and support from the staff of the Division of Biomedical Services, Preclinical Research Facility, University of Leicester, for technical support and care of experimental animals. We thank Carolyn Jones, Lucia Pinon and staff of the histology facility for technical

assistance. We would like to acknowledge the MIND facility (UMC Utrecht Brain Center) for 3D imaging by lightsheet microscopy. This work was supported by funding from the Medical Research Council (L.P.) ITTP (L.P.), Vetenskapsradet (L.P.), Parkinson's funden (L.P.), Pennsylvania Department of Health (SAP #4100042728) (E.S.B., T.A.), 1P50MH096891 (Raquel Gur)–subproject 6773 (E.S.B. and T.A.), Stichting ParkinsonFonds (R.J.P) and Netherlands Organization for Scientific Research (NWO VICI) (R.J.P).

## Author contributions

L.P. conceived and designed the research project, C.S., M.T., K.P., P.G., Y.A., R.V.S, T.O., D.D., E.M.G., R.J.P. and L.P. performed experiments, collected and analyzed the data, E.B., B.P.B., S.L.F., T.A. and E.S.B. contributed new reagents, L.P. wrote the paper. All authors provided critical comments on the manuscript and results.

## Competing interests

The authors declare no competing interests.
