## [Peer Review File · Nature Communications]

Reviewers' Comments:

Reviewer #1:

Remarks to the Author:

The manuscript by Soleilhavoup et al characterizes the role of the TF Nolz1 in striatal neuron fate specification as well as DA axon innervation, adding to a growing list of TFs contributing to striatal neuron development. Key phenotypes of the mutant include defective DA projections, as well as a fate switch from striatonigral to striatopallidal neurons in the striatum. They also reveal mechanisms underpinning the interesting DA projection phenotype including a possible novel role for TGF alpha signaling; this finding could have implications for stem cell derived transplantation methods. The key findings are robust, sufficiently novel, and overall include a substantial body of work. However, a few key points need to be addressed - below are my main points of critique.

1. The authors should explain Darpp32 staining in Fig 2; it looks more clustered, but also more intense in the mutant embryo. The authors should more clearly describe the result.
2. The authors should show Nolz1 expression in embryo more clearly. It is puzzling that Nolz1 is not expressed in the VZ, and yet, birthdating changes are observed. I cannot reconcile/understand non-cell autonomous effects of Nolz1 on neurogenesis. Can the authors explain this better.
3. Fig 3F – could the authors show TGFalpha expression
4. Fig 4F, show higher mag panels
5. Fig 5, GlycoDat in B, D, E does not look similar. This is fundamental to the claim that the phenotype originates in the striatum. Based on these pictures, one could argue that there is a partial phenotype in the En1-cre and Foxd1-cre cKOs (particularly in light of Panmans previous paper showing Nolz1 expression in a subset of DA neurons). An alternative would be to provide some in situs in the En1 cKO showing no striatal marker changes.
6. Fig 6, Tag1 phenotype is unconvincing; in general it seems like this figure belongs in the supplement.
7. It is unclear if the thalamocortical phenotype is cell autonomous or not. The authors should clarify this.
8. "indicating that the establishment of striatonigral projections is regulated cell autonomously by Nolz1."....the authors earlier had claimed the Nolz1 dependent fate switch was non autonomous.....this sentence/concept is confusing
9. The Foxg1-cre cKO should be analyzed for TH, DARP32 at later postnatal days when the anatomy of the striatum, accumbens etc is easier to visualize on coronal sections. Related to this, the authors could comment if these mice have a behavioral phenotype.
10. In Fig 7, in some panels the neurites are inadvertently cropped; this could affect measurements. This should be rectified, and please clarify that measurements were not performed with cropped images.
11. Fig 8, some brightfield panels are not sharp. In 8F and I, an important finding, it is hard to see the starter cells.
12. In the text list Figs in order of appearance; Alpha spelled wrong in last Fig; GlycoDat font is inconsistent.

Reviewer #2:

Remarks to the Author:

In this manuscript, Soleihavoup and colleagues systematically characterized the developmental phenotype of *Nolz1*^{-/-} mutant mouse embryos. *Nolz1*^{-/-} is a transcriptional factor, which has been shown to regulate DA neuronal lineage and striatal projection neurons. Here, the authors showed several interesting findings: 1. In *Nolz1*^{-/-} embryos, DA axons are misguided, cross midline in hypothalamus, and fail to innervate the striatum. 2. *Nolz1*^{-/-} striatum, and striatonigral neuron fate is switched to striatopallidal cell subtype. 3. Using *Nolz1* conditional deletion mice, striatal specific deletion of *Nolz1* could partially recapitulate the DA axon misguidance phenotype. 4. The authors further performed transcriptomic analysis and in vitro assays and proposed candidate axonal guidance cues for the mechanism. Overall, the data is of high quality, and the phenotype is clearly demonstrated. However, the results are primarily descriptive. There is a lack of connection between experiments. In addition, the conclusion is speculative, leaving many questions unanswered. Therefore, the reviewer feels that the manuscript, at least in the current form, is short of the significance and definitive conclusion to be published in the journal. The major points are listed below:

1. Lack of logical connection between DA axon misguidance phenotype and striatal neuron fate phenotype. Using constitutive *Nolz1*^{-/-} mice, the authors found these two striking phenotypes, each of which would be interesting and worth digging deeper. It is not clear whether these two phenotypes are related. *Nolz1*^{-/-} is a transcriptional factor, which may lead to many changes in different cell types. Therefore, it is critical to use cell-type specific approaches to dissect the detailed mechanism.
2. It is striking that DA neuron axons are misguided and crossed the midline in the hypothalamus. But the authors did not try to answer such an important question "Why cross the midline?" (Figure 1B,D). Instead, the authors try to conclude that "the absence of *Nolz1* renders the striatum impermissible for DA fibre innervation". The DA axons cross the midline way before reaching the striatum, this phenotype must have occurred before reaching striatum. In addition, simple conclusion like this is not rigorous. There are other regions that lack the expression of *Nolz1* where DA axons pass through without sprouting.
3. It is very interesting that striatonigral projection neurons are specifically affected by *Nolz1* deletion. Striatonigral and striatopallidal both express *Nolz1*, but only striatonigral neuron fate is changed. This finding is very interesting. The authors should expand this section, examine the mechanism and develop this into an independent paper. The authors should carefully characterize the striatonigral projection, whether they have collaterals in the GPe, and whether they maintain their direct projection to other regions such as GPi, SNr, etc.
4. Using conditional deletion of *Nolz1*, the authors found that *FoxG1*-IRES-Cre;*Nolz1*^{fl/fl} mutant embryos displayed similar phenotypic alterations as observed in *Nolz1*^{-/-} mutant embryos. This is one of the strengths of this paper. However, this should not just to confirm the findings seen in *Nolz1*^{-/-} mice, the authors should have used this as a model for further mechanistic study. Also, it is important to show when the Cre is turned on in the *En1*-Cre, *FoxD1*-Cre, and *FoxG1*-IRES-Cre mouse lines. The DA axons in *Nolz1*^{-/-} mutant embryos crossed the midline, the authors implied that this phenotype may be caused by loss of *Nolz1* in hypothalamus. Did *FoxD1*-Cre;*Nolz1*^{fl/fl} mice show similar midline crossing phenotype?
5. Lastly, the authors used in vitro assays to examine the DA axon innervation of striatum and found that *Tgfa*, *Egfr*, *ErbB4*, *Sema3a* are important. The function of these axonal guidance cues has been extensively studied in many in vitro and in vivo systems. To reach the conclusion that these are the potential mechanism for DA mis-guidance seen in *Nolz1*, the authors should perform in vivo experiments where ligand and receptors are conditionally deleted in either striatum or DA neurons to carefully examine the causal relationship.

Reviewer #3:

Remarks to the Author:

In this study, Soleilhavoup et al showed the requirement of the transcription factor Nolz1 for the correct specification of striatal projection neurons and its effect on the establishment of afferent striatal innervation originating from dopamine neurons in the midbrain. They found that in E18.5 embryos of Nolz1^{-/-} mice, axons from dopaminergic neurons in the midbrain fail to reach their striatal target, with a subset crossing the midline instead, while also observing defective striatonigral projections. Using RNA sequencing and histological analysis they observed gene expression changes associated with striatonigral projections neurons at the expense of genes and markers associated to striatopallidal projection neurons, which suggests a role for Nolz1^{-/-} in controlling the specification of those subtypes. Using conditional knockout models for the selective ablation of the Nolz1 gene in different regions of the brain, they confirm the role of striatum for the defective axonal outgrowth observed. Finally, in experiments using explants of midbrain, striatum and other brain regions from WT and knockout mice, they argue for the role of factors released in the mutant striatum, in particular Tgfa, as the effector of defective dopaminergic axonal outgrowth.

The topic of the manuscript, how neurons of the brain are specified and guided to their appropriate targets is of great interest as the vast majority of axon guidance studies focused on spinal cord development. This manuscript has thus a high potential to impact the field. The experiments performed are of high quality, and use distinct technological approaches to investigate specific points. The article is in general well-written and the main claim important.

However, I believe that the present study must address several major concerns before being acceptable for publication in Nature Communications.

In page 6 the author report that axon phenotype observed in Nolz1 KO is not due to elimination of cells but in Fig S1B it seems that TH and Aldh1a cells are less numerous in KO. Stereological neuron counting should be performed at least for DA markers.

In page 8 Selection criteria for the 61 identified genes from mRNA seq should be defined.

Gad1 is expressed by both the striatopallidal and striatonigral neurons and the authors should not consider Gad1 as a specific gene expressed by striatopallidal neurons.

Fig 3, why the authors did the gene expression validation at E15.5 when the RNA seq was performed at E18.5?

P8, The authors state that the fate change of Nolz1 might be partial since Penk expression is impaired. This is not clear and should be clarified.

P11: The author report that they did not observed any alteration in gene expression in midbrain and hypothalamus of Nolz1 KO but they don't show any results or cite any studies. To claim this point they should show mRNA seq data from laser capture microdissection of DA neurons or midbrain dissection from En-cre Nolz1 flox mice.

Fig 5 Patterns of DA innervation in the striatum appear to be different in the mutant than in the control, especially in respect to striatal patches and matrix. The authors should clarify this point. In mDA neurons, Nolz1 is mainly expressed by VTA neurons. They authors should provide a more detailed analysis of conditional mutant mice (En-cre) and look more closely in nucleus accumbens and mPFC where VTA neurons are mainly projecting to. The author cannot simply show a partial striatal view at one embryonic time point.

It is not clear if some midbrain crossing dopamine axons are present in En-Cre Nolz1 Flox embryos. 3D light-sheet images of these embryos should be analyzed.

Quantification GlycoDAT fibers reaching striatum in different Cre lines should be provided (Fig. 5B-E)

P 11: Why Foxg1Cre Nolz1 Flox mice die one month after birth?
Why is the phenotype of Nolz1^{-/-} less evident than Foxg1Cre?
The authors should discuss it.

Fig 6 P-S cCasp3 appears to be present in axons and axonal cCasp3 have been suggested to be involved in axon targeting. The authors should discuss how ectopic cCasp3 in axons could contribute to the phenotype observed.

Fig. 7AB shows striatal explants and the axonal length of explant from KO is shorter. The authors concluded that Nolz1 regulates the establishment of the striatonigral projections. The authors should rather state that Nolz1 regulates the establishment of striatal projections as no specific marker for striatal subsets were analyzed.

P14 More details should be added for Pcdh10 KO. Where is it expressed? Is it specific to striatum?

Fig. 7 shows explants but the picture is a tile images and explants are not shown entirely. Also, the distance between explants also appears to be different between conditions (WT striatum and KO midbrain are closer to each other than KO striatum and KO midbrain). This should be controlled as distance between explants can have a significant impact on their response. The mean distance between explants for each condition should be quantified and reported. The authors also provide more convincing examples especially for midbrain axons "repelled" by Nolz1 KO striatum.

Explain why is Nolz KO thalamic explants repelled by WT striatal explants (Fig. S5C)?

Figure 8: They used microfluidic device to grow primary DA neurons. How much of the axons shown are dopamine axons? They show b-tubulin positive axons but TH or DAT staining should be analyzed and shown in order to claim that the reactive axons are dopaminergic.

Sema3a appears as a strong candidate for inhibiting DA axon outgrowth. And Tgfa not significant in growth cone collapse test (Fig. 8B). Clarify the choice for Tgfa. And why not test Sema3a in the explant test?

Figure 8 I, what is the control compound for the pharmacological inhibitor?

P 19, the authors claim that Ebf1 KO embryos did not have any defect on the establishment of DA connectivity. However, a study by Yin 2009 has shown that Ebf1 KO display dopaminergic axon defect and SNc dopamine neurons do not innervate the dorsal striatum. This stretch the importance that striatal patterning defect does have an impact on DA connectivity. It is possible that axon defect in Ebf1 KO comes from DA neurons themselves as they express Ebf1 during a critical period.

Minors:

The figure should be organized such as they are presented as cited in the text. It is quite annoying jumping from one fig to the other while reading.

For mRNA sequencing data, the number of differentially expressed gene and the criteria for the cutoff should be indicated.

Arrowhead missing Fig 1B

P6: use (Fig1Q-T) instead of (1Q-T)

P8: Reference for Ebf1 expressed by striatonigral neurons should be provided.

P8: last sentence, a period is missing.

Explain why use GlycoDAT and other axonal markers in results section.

Figure 4A - specify colors of staining

Fig 6U and 6V are missing.

Fig 7C-F should be shown along with DAPI or other markers since it is difficult to figure what the images are showing.

Correct "Fig 7D,E" in page 14 to "Fig 7E,F"

In Fig S2C the difference in Isl1+ cells are not that evident. Quantifications needed.

Response to reviewers

We thank all three reviewers for very insightful and helpful comments that significantly improved our manuscript. As outlined below we have provided new data in figure 2, 5, 7, S1, S3, S4 and S5. In addition, we have improved several images and reorganized the figures as requested by the reviewers. Furthermore, we have made several alterations to the text to clarify reviewers' remarks and describe the new data. We hope that the reviewers agree that we now provide sufficient novel insight into the role of Nolz1 in striatal development and dopaminergic axon guidance that is suitable for publication in Nature Communications.

Reviewer 1

The manuscript by Soleilhavoup et al characterizes the role of the TF Nolz1 in striatal neuron fate specification as well as DA axon innervation, adding to a growing list of TFs contributing to striatal neuron development. Key phenotypes of the mutant include defective DA projections, as well as a fate switch from striatonigral to striatopallidal neurons in the striatum. They also reveal mechanisms underpinning the interesting DA projection phenotype including a possible novel role for TGF alpha signaling; this finding could have implications for stem cell derived transplantation methods. The key findings are robust, sufficiently novel, and overall include a substantial body of work. However, a few key points need to be addressed - below are my main points of critique.

1. The authors should explain Darpp32 staining in Fig 2; it looks more clustered, but also more intense in the mutant embryo. The authors should more clearly describe the result.

As the reviewer pointed out Darpp32 stain is indeed more clustered in Nolz1^{-/-} mutant embryos. We have analysed this in more detail in the revised version. From the analysis of Darpp32 in combination with other markers including L1, Ctip2 and bGal we found that these clusters represents Darpp32⁺L1⁺ labelled axonal tracts from the cortex, which are abnormally fasciculated within the striatum. These clusters are devoid of Ctip2 and bGal expression further demonstrating that Darpp32 is not expressed by projection neurons in the Nolz1^{-/-} mutant striatum, in contrast to wild-type embryos. Furthermore, RNA sequencing analysis revealed that Darpp32 expression is downregulated in the mutant striatum compared to the wild-type. Based on our analysis we do not believe that the intensity of Darpp32 expression in the mutant embryos is higher compared to the wild-type. We have included the new expression data in Fig. 2 and described these results in more detail at page 7.

2. The authors should show Nolz1 expression in embryo more clearly. It is puzzling that Nolz1 is not expressed in the VZ, and yet, birthdating changes are observed. I cannot

reconcile/understand non-cell autonomous effects of Nolz1 on neurogenesis. Can the authors explain this better.

We show now the expression of Nolz1 in the E11.5 striatum in figure S4J. The expression analysis of Nolz1 at E11.5 indeed shows that Nolz1 is not expressed in the ventricular zone, but is confined to the subventricular zone. The exclusion of Nolz1 from proliferating cells in the ventricular and subventricular zone has also been shown by Ko et al., 2013; Neuroscience Letters. In the absence of Rarb, which is predominantly expressed in post-mitotic neurons in the striatum, there is a reduction in the proliferation within the ventricular zone. In that case the effect on the proliferation is mediated by Fgf3 (Rataj-Baniowska et al., 2015; Journal of Neuroscience). Nolz1 could regulate proliferation in a similar manner, through the induction of other secreted proteins that impact on the ventricular zone. We explain this in further detail in the discussion of the revised version at page 18.

3. Fig 3F – could the authors show TGFA α expression

We have included Tgfa expression in the graph of Fig. 3F.

4. Fig 4F, show higher mag panels

We now show higher magnification panels in figure 4F.

5. Fig 5, GlycoDat in B, D, E does not look similar. This is fundamental to the claim that the phenotype originates in the striatum. Based on these pictures, one could argue that there is a partial phenotype in the En1-cre and Foxd1-cre cKOs (particularly in light of Panmans previous paper showing Nolz1 expression in a subset of DA neurons). An alternative would be to provide some in situs in the En1 cKO showing no striatal marker changes.

We agree with the reviewer that the images in 5B, D and E would suggest a reduction of striatal innervation in EnCre and FoxD1 line. However, the observed differences are mainly due to differences in the level of the striatum from which these images were taken from. We have analysed several mutant brains of the different Cre lines and can confirm that the innervation by GlycoDat in the FoxD1Cre;Nolz1 and EnCre;Nolz1 is similar as to the controls. In the revised version we show images that better represent the phenotype. In addition, we have measured the striatal fibre density using ImageJ, further demonstrating that there is no decrease in striatal innervation when Nolz1 is selectively ablated in either the midbrain (EnCre) or the hypothalamus (FoxD1Cre). The new data can be found in Figure 5G. As suggested by the reviewer, we have also analysed several striatal markers in the EnCreNolz1 brain and as expected there were no changes in striatal gene expression. These data can be found in Fig. S5I and is mentioned in the text at page 12.

6. Fig 6, Tag1 phenotype is unconvincing; in general it seems like this figure belongs in the supplement.

We reassessed the Tag1 phenotype and agree that the changes are very mild. Therefore, we have removed this data from the manuscript. Furthermore, we have moved the data presented in Figure 6 to the supplemental figure 6, except for the Dil data, which is in Fig. 7a now.

7. It is unclear if the thalamocortical phenotype is cell autonomous or not. The authors should clarify this.

Forebrain selective ablation of Nolz1 using the FoxG1IresCre line results in abnormal thalamocortical projections similar to Nolz1 null mutants. There, we conclude that the thalamocortical phenotype is cell non-autonomous and due to striatal patterning abnormalities. We describe this in the text at page 13 as follows: "The phenotypic resemblance between the constitutive and conditional striatal-specific Nolz1 mutant mouse lines demonstrates striatal, non-cell autonomous requirement of Nolz1 in orchestrating the attraction and guidance of DA and other axonal tracts through the striatum".

8. "indicating that the establishment of striatonigral projections is regulated cell autonomously by Nolz1."....the authors earlier had claimed the Nolz1 dependent fate switch was non autonomous.....this sentence/concept is confusing

We agree with the reviewer that this concept is confusing. Therefore we altered the sentence on page 13 as follows: "Consistent with the absence of striatonigral projections, the axonal length in striatal explants from Nolz1^{-/-} mutant embryos was significantly shorter compared to wild-type explants (Fig. 6E, F)".

9. The Foxg1-cre cKO should be analyzed for TH, DARP32 at later postnatal days when the anatomy of the striatum, accumbens etc is easier to visualize on coronal sections. Related to this, the authors could comment if these mice have a behavioral phenotype.

It has not been possible to analyse the phenotype and behavioural consequences in the FoxG1IresCre;Nolz1 line as the pups almost immediately die after birth. We have mentioned this at page 12.

10. In Fig 7, in some panels the neurites are inadvertently cropped; this could affect measurements. This should be rectified, and please clarify that measurements were not performed with cropped images.

We have replaced the images for uncropped images and confirm that the measurements were not taken from the cropped images.

11. Fig 8, some brightfield panels are not sharp. In 8F and I, an important finding, it is hard to see the starter cells.

We have replaced some of the brightfield images that were not sharp enough. We have further increased the intensity in 8F and I so the axonal projections are better visible.

12. In the text list Figs in order of appearance; Alpha spelled wrong in last Fig; GlycoDat font is inconsistent.

We have changed the order of the figures and removed inconsistencies and spelling errors.

Reviewer 2

In this manuscript, Soleihavoup and colleagues systematically characterized the developmental phenotype of *Nolz1*^{-/-} mutant mouse embryos. *Nolz1*^{-/-} is a transcriptional factor, which has been shown to regulate DA neuronal lineage and striatal projection neurons. Here, the authors showed several interesting findings: 1. In *Nolz1*^{-/-} embryos, DA axons are misguided, cross midline in hypothalamus, and fail to innervate the striatum. 2. *Nolz1*^{-/-} striatum, and striatonigral neuron fate is switched to striatopallidal cell subtype. 3. Using *Nolz1* conditional deletion mice, striatal specific deletion of *Nolz1* could partially recapitulate the DA axon misguidance phenotype. 4. The authors further performed transcriptomic analysis and in vitro assays and proposed candidate axonal guidance cues for the mechanism. Overall, the data is of high quality, and the phenotype is clearly demonstrated. However, the results are primarily descriptive. There is a lacking of connection between experiments. In addition, the conclusion is speculative, leaving many questions unanswered. Therefore, the reviewer feels that the manuscript, at least in the current form, is short of the significance and definitive conclusion to be published in the journal. The major points are listed below:

1. Lack of logical connection between DA axon misguidance phenotype and striatal neuron fate phenotype. Using constitutive *Nolz1*^{-/-} mice, the authors found these two striking phenotypes, each of which would be interesting and worth digging deeper. It is not clear whether these two phenotypes are related. *Nolz1*^{-/-} is a transcriptional factor, which may lead to many changes in different cell types. Therefore, it is critical to use cell-type specific approaches to dissect the detailed mechanism.

*In the manuscript we provide several lines of evidence that there is clear link between the striatal patterning defects observed in *Nolz1* mutant embryos and axon guidance defects. Using a conditional gene ablation approach we showed that striatal selective ablation of *Nolz1* results in a similar phenotype as observed in *Nolz1*^{-/-} mutant embryos with regard to DA axonal innervation and guidance. Also other axonal forebrain tracts were similarly effected in both *FoxG1IRES-CreNolz1^{fl/fl}* conditional and *Nolz1*^{-/-} mutant embryos. In addition, we found evidence that factors secreted from the *Nolz1*^{-/-} mutant striatum imposes*

a repulsive effect on DA axons, which can be restored by the addition signalling molecules normally expressed in the striatum. Striatal patterning defects also causes lack of striatal axonal outgrowth. However, from our analysis we could conclude that the guidance phenotype is caused by altered composition of secreted factors and not due to the lack of striatal axonal extensions. Furthermore, selective ablation of Nolz1 in other relevant brain regions using the FoxD1Cre (hypothalamus) and En1Cre lines did not observe in any guidance defects.

2. It is striking that DA neuron axons are misguided and crossed the midline in the hypothalamus. But the authors did not try to answer such an important question “Why cross the midline?” (Figure 1B,D). Instead, the authors try to conclude that “the absence of Nolz1 renders the striatum impermissible for DA fibre innervation”. The DA axons cross the midline way before reaching the striatum, this phenotype must have occurred before reaching striatum. In addition, simple conclusion like this is not rigorous. There are other regions that lacks the expression of Nolz1 where DA axons pass through without sprouting.

From our analysis of FoxG1IRESCreNolz1fl/fl mutant embryos in which Nolz1 is selectively ablated in the striatum we conclude that striatal Nolz1 expression is both required for normal axonal pathfinding in the hypothalamus and striatal innervation. The absence of striatal Nolz1 expression does not only renders the striatum impermissible for DA fibre innervation, but also causes midline crossing. We do not claim that the lack of striatal innervation causes the remaining axons to cross the midline. Instead, we provide evidence that the altered secretion of factors from the striatum causes midline crossing of a subset of DA axons. Furthermore, we have analysed the sequence of events in Nolz1 mutant embryos and found that at E13.5 a subset of DA axons has already reached the striatum, while midline crossing could not be observed at that time yet. Although it is not understood why DA axons cross the midline, it is clear from our analysis that DA axonal misguidance is caused by the striatal phenotype.

3. It is very interesting that striatonigral projection neurons are specifically affected by Nolz1 deletion. Striatonigral and striatopallidal both express Nolz1, but only striatonigral neuron fate is changed. This finding is very interesting. The authors should expand this session, examine the mechanism and develop this into an independent paper. The authors should carefully characterize the striatonigral projection, whether they have collaterals in the GPe, and whether they maintain their direct projection to other regions such as GPi, SNr, etc.

In Nolz1 mutant embryos striatonigral neurons are not formed and hence the striatonigral projections are lost. The innervation of SNr by striatonigral projections was lost as indicated by the lack of Darpp32 expression in the midbrain (Fig. 2O-T). Also, Dil injections in the striatum revealed a loss of striatonigral projections towards the midbrain (Fig. 6A-D).

4. Using conditional deletion of Nolz1, the authors found that FoxG1-IRES-Cre;Nolz1fl/fl mutant embryos displayed similar phenotypic alterations as observed in Nolz1^{-/-} mutant

embryos. This is one of the strengths of this paper. However, this should not just to confirm the findings seen in *Nolz1*^{-/-} mice, the authors should have used this as a model for further mechanistic study. Also, it is important to show when the Cre is turned on in the *En1*-Cre, *FoxD1*-Cre, and *FoxG1*-IRES-Cre mouse lines. The DA axons in *Nolz1*^{-/-} mutant embryos crossed the midline, the authors implied that this phenotype may be caused by loss of *Nolz1* in hypothalamus. Did *FoxD1*-Cre;*Nolz1*^{fl/fl} mice show similar midline crossing phenotype?

In the manuscript we follow up on the striatal specific requirement of Nolz1 for DA axon guidance. We show that the Nolz1 mutant striatum exerts a repulsive effect on DA axons and we identify Tgfa as a novel striatum expressed chemoattractant for DA axons. We provide now a more detailed analysis of the conditional mouse lines and show that the Nolz1 allele has already been recombined by E11.5 (Fig. S5A). This is described at page 11. Because of the Nolz1 expression in the hypothalamus we hypothesized that the phenotype could be caused by the loss of hypothalamic Nolz1 expression. In the text we wrote: Although we did not observe any alteration in gene expression in the midbrain and hypothalamus of Nolz1^{-/-} mutant embryos, the involvement of Nolz1 in regulating DA axon guidance in these regions could not be totally ruled out. To investigate this further we generated conditional mutant mouse lines in which Nolz1 was selectively ablated from either the midbrain, hypothalamus or striatum. As shown in Figure 5D and S5G there is no midline crossing observed in FoxD1Cre;Nolz1^{fl/fl} mutant embryos

5. Lastly, the authors used in vitro assays to examine the DA axon innervation of striatum and found that *Tgfa*, *Egfr*, *ErbB4*, *Sema3a* are important. The function of these axonal guidance cues has been extensively studied in many in vitro and in vivo systems. To reach the conclusion that these are the potential mechanism for DA mis-guidance seen in *Nolz1*, the authors should perform in vivo experiments where ligand and receptors are conditionally deleted in either striatum or DA neurons to carefully examine the causal relationship.

From the in vitro analysis we found that the Nolz1^{-/-} mutant striatum has a repulsive effect on DA axonal outgrowth. We identified Tgfa and Sema3a as genes downregulated in the Nolz1^{-/-} mutant striatum and showed that Tgfa could rescue the Nolz1^{-/-} mutant phenotype. From our analysis we concluded that the addition of Tgfa is sufficient to promote DA axonal outgrowth. We do not claim that Tgfa signalling from the striatum is required for DA axon guidance and we altered the last sentence of the results section at page 16 as follows: "These data demonstrate that Tgfa signalling activation is sufficient to attract DA axonal projections".

Instead we think that a combination of several factors secreted from the striatum are involved in DA axon guidance. In the discussion we wrote that the repulsive environment from the Nolz1^{-/-} mutant striatum is caused by the altered expression of several secreted factors. Tgfa, Egfr, ErbB4 and Sema3a mutant embryos do not show a DA axon guidance phenotype (Blum et al., 1998; Nature Neuroscience, Torre et al., 2010; Mol. Cell. Neurosci., Thuret et al., 2004 Journal of Neurochemistry, Kornblum et al., 1998 J. Neurosci. Res.), further suggesting that several pathways might be involved. Therefore, the generation and analysis of conditional

mouse lines will not provide any further information. Nevertheless, the implication of Tgfa in promoting DA axonal outgrowth is a novel finding and has potential for future transplantation studies.

Reviewer 3

In this study, Soleilhavoup et al showed the requirement of the transcription factor Nolz1 for the correct specification of striatal projection neurons and its effect on the establishment of afferent striatal innervation originating from dopamine neurons in the midbrain. They found that in E18.5 embryos of Nolz1^{-/-} mice, axons from dopaminergic neurons in the midbrain fail to reach their striatal target, with a subset crossing the midline instead, while also observing defective striatonigral projections. Using RNA sequencing and histological analysis they observed gene expression changes associated with striatonigral projection neurons at the expense of genes and markers associated to striatopallidal projection neurons, which suggests a role for Nolz1^{-/-} in controlling the specification of those subtypes. Using conditional knockout models for the selective ablation of the Nolz1 gene in different regions of the brain, they confirm the role of striatum for the defective axonal outgrowth observed. Finally, in experiments using explants of midbrain, striatum and other brain regions from WT and knockout mice, they argue for the role of factors released in the mutant striatum, in particular Tgfa, as the effector of defective dopaminergic axonal outgrowth.

The topic of the manuscript, how neurons of the brain are specified and guided to their appropriate targets is of great interest as the vast majority of axon guidance studies focused on spinal cord development. This manuscript has thus a high potential to impact the field. The experiments performed are of high quality, and use distinct technological approaches to investigate specific points. The article is in general well-written and the main claim important.

However, I believe that the present study must address several major concerns before being acceptable for publication in Nature Communications.

1. In page 6 the author report that axon phenotype observed in Nolz1 KO is not due to elimination of cells but in Fig S1B it seems that TH and Aldh1a cells are less numerous in KO. Stereological neuron counting should be performed at least for DA markers.

From the images it may seem that there might be a slight reduction in the number of TH and Aldh1a1 positive neurons. We have carefully reanalysed several embryos (n=3) that were stained for Aldh1a1 and TH and quantified the percentage Aldh1a1 labelled TH⁺ neurons in each fourth section of each embryo. We did not observe any reduction in the number of TH and Aldh1a1 expressing neurons in the mutant midbrain. We show these results in figure S1D,

E. and describe the methodology at page 48. Therefore, we think that the slight reduction in Aldh1a1 expressing neurons that might be observed in Fig. S1C is mainly caused by a difference in anatomical level between the wild-type and mutant embryo the images were taken from.

2. In page 8 Selection criteria for the 61 identified genes from mRNA seq should be defined.

We have at randomly selected 61 genes from a list of differentially expressed genes, which we further verified by in situ hybridization. We have modified the text at page 8 accordingly.

3. Gad1 is expressed by both the striatopallidal and striatonigral neurons and the authors should not consider Gad1 as a specific gene expressed by striatopallidal neurons.

We have mistakenly considered Gad1 as a pallidal selective gene. We have changed figure 3 and the text accordingly.

4. Fig 3, why the authors did the gene expression validation at E15.5 when the RNA seq was performed at E18.5?

The reviewer asked why the verification was carried out in E15.5 embryos while the RNA seq data was obtained from E18.5 embryos. Since nigral and pallidal lineages are already established at E15.5, we decided to use E15.5 embryos for in situ validation. For completion we have analysed some of the differentially expressed genes in E18.5 embryos (data not shown) and we obtained the same results (see page 8).

5. The authors state that the fate change of Nolz1 might be partial since Penk expression is impaired. This is not clear and should be clarified.

The description at page 8 was indeed not totally clear. Pallidal genes, including Penk are upregulated in Nolz1^{-/-} mutant embryos and our data suggests that there is a nigral to pallidal lineage switch. We also analysed whether the pallidal neurons innervate their target, the globus pallidus, in Nolz1 mutant embryos. To analyse pallidal axonal projections, we used Penk as a marker. While Penk expression is upregulated in Nolz1^{-/-} mutant striatum, the innervation of the globus pallidus by Penk labelled axons is impaired. We have changed the sentence in the text as follows: "Despite the upregulation of several pallidal markers, the projections of striatopallidal neurons towards the GP were impaired in Nolz1^{-/-} mutant embryos as revealed by the analysis of Penk (Fig. S2E)".

6. P11: The author report that they did not observed any alteration in gene expression in midbrain and hypothalamus of Nolz1 KO but they don't show any results or cite any studies. To claim this point they should show mRNA seq data from laser capture microdissection of DA neurons or midbrain dissection from En-cre Nolz1 flox mice.

Since in EnCre;Nolz1fl/fl and Foxd1Cre;Nolz1fl/fl conditional mutant embryos DA axons follow a normal trajectory towards the striatum it is not expected that gene expression changes in the midbrain and/or hypothalamus underlie the axon guidance phenotype in Nolz1 mutant

embryos. Indeed, we did not observe any altered gene expression in midbrain DA (see also Fig. S1) neurons and hypothalamus. In the revised version of the manuscript we show in situ data for some of these genes involved in axon guidance and patterning in the midbrain and hypothalamus of Nolz1^{-/-} mutant embryos. The new data can be found in supplemental figure S3A, B. We have also changed the sentence at page 11 and state that we did not observe any gene expression changes in Nolz1^{-/-} mutant midbrain and hypothalamus that could explain the DA axon guidance phenotype. We agree that we cannot exclude that there will not be any gene expression changes in the Nolz1 mutant midbrain, but from the analysis of the En1Cre conditionally ablated Nolz1 mutant embryos we can conclude those changes most likely will not lead to DA axon guidance defects. To investigate an eventual role of Nolz1 in the midbrain that is unrelated to DA axon guidance would be interesting, but is beyond the scope of this manuscript.

7. Fig 5 Patterns of DA innervation in the striatum appear to be different in the mutant than in the control, especially in respect to striatal patches and matrix. The authors should clarify this point. In mDA neurons, Nolz1 is mainly expressed by VTA neurons. They authors should provide a more detailed analysis of conditional mutant mice (En-cre) and look more closely in nucleus accumbens and mPFC where VTA neurons are mainly projecting to. The author cannot simply show a partial striatal view at one embryonic time point.

The striatal innervation in Figure 5 seems different, but that appeared to have mainly been due to differences in striatal level from which the images were taken from. We have now carefully analysed several mutant embryos and chosen images that better represent the phenotype. According to the new data there are no differences in the pattern of striatal innervation between wild-type, En1Cre and FoxD1Cre conditionally ablated Nolz1 mutant embryos. We have quantified striatal DA fibre density, results that can be found in Figure 5G, and we can confirm that there are no differences between the control, En1Cre and FoxD1Cre Nolz1 ablated embryos. Striatal patterning is also normal in EnCre;Nolz1^{fl/fl} mutant embryos as can be seen in Figure S5I of the revised version. However, the striosome and matrix organization is disrupted in constitutive Nolz1^{-/-} mutant embryos. We show Mor and Calbindin expression data in supplemental figure S3H,J in the revised version and describe the new data at page 10. We have also more carefully analysed En1Cre;Nolz1^{fl/fl} mutant embryonic and adult brains and can confirm that DA innervation of the PFC and nucleus accumbens in the conditional mutants. The new data is presented in Fig S5D-F and is described in the text at page 12.

8. It is not clear if some midbrain crossing dopamine axons are present in En-Cre Nolz1 Flox embryos. 3D light-sheet images of these embryos should be analyzed.

The reviewer wants to know whether there is any midline crossing of dopaminergic axons in EnCre;Nolz1^{fl/fl} mutant embryos. We can confirm that there is no midline crossing of dopaminergic axons in EnCre;Nolz1^{fl/fl} conditional mutant embryos. We have analysed several mutant embryos and taken images of several levels throughout the hypothalamus. These images can be found in fig. S5G, H.

9. Quantification GlycoDAT fibers reaching striatum in different Cre lines should be provided (Fig. 5B-E)

The reviewer asked to measure the striatal dopaminergic fibre density in the different conditional mouse lines. We have measured the GlycoDat⁺ fibre density in the striatum of E18.5 control, Nolz1^{-/-}, EnCre; Nolz1^{fl/fl}, FoxD1Cre; Nolz1^{fl/fl} and FoxG1IresCre; Nolz1^{fl/fl} mutant embryos and found that the striatal innervation in EnCre; Nolz1^{fl/fl} and FoxD1Cre; Nolz1^{fl/fl} conditional mutant embryos was similar to the controls. However, in Nolz1^{-/-} and FoxG1IresCre; Nolz1^{fl/fl} the fibre density was significantly reduced compared to the control embryos. The graph showing these results can be found in Figure 5G of the revised version. The methodology is described at page 48.

10. P 11: Why Foxg1Cre Nolz1 Flox mice die one month after birth?

Why is the phenotype of Nolz1^{-/-} less evident than Foxg1Cre?

The authors should discuss it.

The connectivity between the cortex and thalamus in FoxG1IresCre;Nolz1^{fl/fl} conditional mutant embryos is severely impaired, which may result in the pups dying within a couple of days after birth. But we do not know for sure why the conditional animals die. We have changed the sentence at page 12 into the following: "FoxG1IresCre;Nolz1^{fl/fl} conditional mutant mice die within one month after birth for unknown reasons". Nolz1^{-/-} mutant mice die immediately after birth, which could be due to a potential the role of Nolz1 in lung development.

The Foxg1IresCreNolz1^{fl/fl} phenotype is milder compared to Nolz1 mutant phenotype. Since Foxg1IresCreNolz1^{fl/fl} mutant embryos exhibit midline crossing and lack of striatal innervation it demonstrates the involvement of the striatum in DA axon guidance. The milder phenotype could be due to differences in the allelic design between the constitutive and conditional Nolz1 mutant mouse line. The constitutive Nolz1 knockout allele comprises the coding region in exon2, intron 2 and exon3, while in the conditional allele only exon 3 is ablated (see figure 5A and S1A). The phenotype of CmvCre;Nolz1^{fl/fl} mutant embryos is also less strong compared to Nolz1^{-/-} null mutant embryos. Another explanation for the phenotypic difference is that other Nolz1 expression regions besides the striatum, hypothalamus and midbrain are involved in DA axon guidance. We have included this in the discussion at page 20 of the revised version.

11. Fig 6 P-S cCasp3 appears to be present in axons and axonal cCasp3 have been suggested to be involved in axon targeting. The authors should discuss how ectopic cCasp3 in axons could contribute to the phenotype observed.

We observe an induction of cCasp3 selectively in thalamic axons that project towards the striatum. We hypothesised that the induction of cCasp3 is caused by a lack of trophic factors that are normally emanated from the cortex. However, it might indeed be possible that the upregulation of cCasp3 itself causes axon guidance defects as it has been shown that Caspases influences responses to axon guidance molecules. We describe this further in the discussion at page 21 and cite Newquist et al., 2013 Cell Reports in the revised version of the manuscript.

12. Fig. 7AB shows striatal explants and the axonal length of explant from KO is shorter. The authors concluded that Nolz1 regulates the establishment of the striatonigral projections. The authors should rather state that Nolz1 regulates the establishment of striatal projections as no specific marker for striatal subsets were analyzed.

We have indeed not labelled the striatal explants for specific striatonigral markers. However, since striatonigral projections are absent in Nolz1 mutant embryos it is expected that this would be the same in the explants. Striatopallidal axonal projections are shorter than nigral projections, which is in line with a shorter outgrowth from the explant. We have changed this sentence at page 13 into: Consistent with the absence of striatonigral projections, the axonal length in striatal explants from Nolz1^{-/-} mutant embryos was significantly shorter compared to wild-type explants (Fig. 7A, B).

13. P14 More details should be added for Pcdh10 KO. Where is it expressed? Is it specific to striatum?

As described by Uemura et al., 2007; Nature Neuroscience Pcdh10 is initially selectively expressed in the striatum (E13.5) and later also observed in the thalamus, globus pallidus and cortex. At the stage we analysed the mutants (E14.5) Pcdh10 expression was mainly restricted to the striatum and therefore we can conclude that the phenotype is caused by its function in the striatum. Pcdh10 mutant embryos have a lack of striatal outgrowth and exhibit a defect in the establishment of thalamocortical projections. Furthermore, striatal axons fail to innervate the SN in Pcdh10 mutant embryos. We have described the phenotype and expression pattern in further detail at page 14

14. Fig. 7 shows explants but the picture is a tile images and explants are not shown entirely. Also, the distance between explants also appears to be different between conditions (WT striatum and KO midbrain are closer to each other than KO striatum and KO midbrain). This should be controlled as distance between explants can have a significant impact on their response. The mean distance between explants for each condition should be quantified and reported. The authors also provide more convincing examples especially for midbrain axons “repelled “ by Nolz1 KO striatum.

The distance between the explants is indeed an important factor that could influence the experimental outcome. We have measured the distances between the striatal and midbrain explants and can confirm that the distance is equal in all conditions. The measurements are displayed in a graph in Fig. 6N in the revised version of the manuscript. We have also improved the images in Figure 6K and show the explants entirely.

15. Explain why is Nolz KO thalamic explants repelled by WT striatal explants (Fig. S5C)?

Thalamic Nolz1 mutant explants are indeed repelled by the wild-type striatum. Nolz1 is expressed in the thalamus (Figure S6Q) where it may mediate the attractive response to growth factors emanated from the striatum. We describe this at page 15 of the revised version.

16. Figure 8: They used microfluidic device to grow primary DA neurons. How much of the axons shown are dopamine axons? They show b-tubulin positive axons but TH or DAT staining should be analyzed and shown in order to claim that the reactive axons are dopaminergic.

A fraction of the neurons grown in the microfluidic device express TH. We show this data in Figure S8E, F.

17. Sema3a appears as a strong candidate for inhibiting DA axon outgrowth. And Tgfa not significant in growth cone collapse test (Fig. 8B). Clarify the choice for Tgfa. And why not test Sema3a in the explant test?

Factors secreted from Nolz1 mutant striatum exert a repulsive effect which is either caused by the lack of attractive or upregulation of repulsive factors. Sema3a is indeed repulsive, but is downregulated in Nolz1 mutant striatum. Therefore, it is unlikely that the phenotype is caused by the lack of Sema3a expression. Tgfa is absent in the mutant and therefore it is more likely that the phenotype is mediated by Tgfa

18. Figure 8 I, what is the control compound for the pharmacological inhibitor?

DMSO was used as a control for the pharmacological inhibitor as shown in 8I. The addition of DMSO to the wild-type striatum did not have any effect on the axonal outgrowth. We mention this information in the material and methods section at page 45 now. The addition of DMSO did not have any effect on the axonal outgrowth.

19. P 19, the authors claim that Ebf1 KO embryos did not have any defect on the establishment of DA connectivity. However, a study by Yin 2009 has shown that Ebf1 KO display dopaminergic axon defect and SNc dopamine neurons do not innervate the dorsal stratum. This stretch the importance that striatal patterning defect does have an impact on DA connectivity. It is possible that axon defect in Ebf1 KO comes from DA neurons themselves as they express Ebf1 during a critical period.

The reviewer pointed out that in the study by Yin et al 2009 there is a loss of DA connectivity in Ebf1^{-/-} mutant embryos. In the study by Yin et al there is a reduction of striatal innervation in E16.5 mutant embryos. However, the striatal innervation seems to be restored in P20 Ebf1 mutant animals (Garel et al., 1999). Thus we think that there is a delay in striatal innervation in Ebf1 mutant embryos. It is very like that this is caused by a defect in the lateral migration of SN neurons, however due to the lack of conditional mouse lines the involvement of the striatum cannot be totally ruled out. Therefore, we have changed in sentence at page 19 as follows: "Also, the striatum still gets innervated by DA axons in Ebf1^{-/-} and Rarb^{-/-} mutant embryos in which the specification of striatonigral neurons is impaired". We have also included Yin et al 2009 as a reference now.

Minors:

1. The figure should be organized such as they are presented as cited in the text. It is quite annoying jumping from one fig to the other while reading.

We have reorganized the figures and they are presented as cited in the text now.

2. For mRNA sequencing data, the number of differentially expressed gene and the criteria for the cutoff should be indicated.

The cut-off criteria to determine differentially expressed genes were as follows: "Genes were identified as being differentially expressed at an adjusted p-value of less than 0.05 for the DESeq2 results and with a fold change of at least 1.5". This is described in the Materials and Methods section at page 51. Using these criteria we identified 139 genes that were upregulated in Nolz1 mutant embryos compared to the wild-type, while 166 genes were downregulated.

3. Arrowhead missing Fig 1B

We have added the arrowhead to Fig. 1B pointing to axons that are halted in front of the striatum.

4. P8: Reference for Ebf1 expressed by striatonigral neurons should be provided.

We have added Lobo et al., 2006; Nature Neuroscience as a reference for Ebf1 expression in striatonigral neurons.

5. P8: last sentence, a period is missing.

We did not understand what the reviewer meant by "a period is missing" in the last sentence of page 8.

6. Explain why use GlycoDAT and other axonal markers in results section.

TH is expressed by all midbrain dopaminergic neurons and labels both cell bodies and axonal projections. We used GlycoDat to label a selective population of dopaminergic axons namely from the SN and dorsal part of the VTA. This is mentioned at page 5 in the results section.

7. Figure 4A - specify colors of staining

We have specified the colours in Figure 4a, with Ebf1 in red and Dapi in blue

8. Fig 6U and 6V are missing.

We made a mistake with the figure labelling and we have corrected this now.

9. Fig 7C-F should be shown along with DAPI or other markers since it is difficult to figure what the images are showing.

We have made the layout of figure 6G-J clearer now by increasing the brightness and labelling

10. Correct "Fig 7D,E" in page 14 to "Fig 7E,F"

We have changed the labelling of Fig. 7D,E to Fig. 6I,J at page 14.

11. In Fig S2C the difference in Isl1+ cells are not that evident. Quantifications needed.

We have quantified the number of aberrant localized cells and there is indeed an increased number of Isl1 positive cells in the GP. The results can be found in Figure S2C

Reviewers' Comments:

Reviewer #1:

Remarks to the Author:

My critiques have been addressed in a reasonable manner, and overall this paper is a positive contribution to the field.

Reviewer #2:

Remarks to the Author:

In the previous manuscript, the authors described an interesting phenotype shown in *Nolz1*^{-/-} mice. However, there were many missing links, and errors in figure and labeling, etc. In this resubmission, the manuscript has improved a lot on data reporting. The additional and corrected figure panels are very helpful.

Reviewer #3:

Remarks to the Author:

Overall, the authors did a good job in answering my comments.

However, to my opinion, they should do the following minor corrections before being acceptable for Nat Com.

For my first point, the author performed neuronal count of TH⁺ and Aldh1a⁺ cells (n=3) on embryonic section at e18.5. Although they claim that the number is not different between mutant and control, it seems that there is in fact a reduction. The authors should at least discuss it or tune down the interpretation.

Fig S2e: remove TH from the panel since it is not shown.

Fig S3h: MOR staining is not of good quality. Labelling in h and i should be corrected (*Nolz1*^{-/-} not *Nolz1*^{-/})

Point 7: The authors made a good effort to answer my critic but since DA projections to PFC only start to innervate the PFC at e18.5 it would have been pertinent to show a later time point. I recommend that they tune down their interpretation in page 12.

Point 8: The authors show images at different levels throughout the hypothalamus but the images are truncated at the bottom, so it is not clear if some axons are actually crossing. They should provide the full images.

Minor point 5: check punctuation P9 : ... specification of striatonigral projection neurons and in the absence of *Nolz1* expression several striatopallidal markers are ectopically expressed in the striatum (.)

Response to reviewers

We thank the reviewers for their positive comments. We have made some minor corrections to the manuscript as suggested by reviewer number 3.

Reviewer 1:

My critiques have been addressed in a reasonable manner, and overall this paper is a positive contribution to the field.

Reviewer 2:

In the previous manuscript, the authors described an interesting phenotype shown in Nolz1-/- mice. However, there were many missing links, and errors in figure and labeling, etc. In this resubmission, the manuscript has improved a lot on data reporting. The additional and corrected figure panels are very helpful.

Reviewer 3:

Overall, the authors did a good job in answering my comments. However, to my opinion, they should do the following minor corrections before being acceptable for Nat Com.

For my first point, the author performed neuronal count of TH+ and Aldh1a+ cells (n=3) on embryonic section at e18.5. Although they claim that the number is not different between mutant and control, it seems that there is in fact a reduction. The authors should at least discuss it or tune down the interpretation.

We think that the images of the wild-type and mutant have been taken from slightly different levels of the brain and therefore the expression seems different between wild-type and mutant embryos. We agree with the reviewer that the images do not fully represent what is written in the text and therefore we tuned down the interpretation and changed the sentence as follows: "Analysis of several markers did not reveal striking changes in DA neuron selective marker expression that could explain the defects in DA axon guidance and innervation."

Fig S2e: remove TH from the panel since it is not shown.

We have removed TH from the panel

Fig S3h: MOR staining is not of good quality. Labelling in h and i should be corrected (Nolz1-/- not Nolz1-/-)

We agree with the reviewer that the staining in Fig. S3h is not of a very good quality. However, the image clearly shows that the localization of MOR is altered in the mutant. We have corrected the labelling in Fig. S3h and i.

Point 7: The authors made a good effort to answer my critic but since DA projections to PFC only start to innervate the PFC at e18.5 it would have been pertinent to show a later time point. I recommend that they tune down their interpretation in page 12.

*We agree with the reviewer that later timepoints would be more conclusive. However we feel it is safe to conclude that there are no differences in the early phases of PFC innervation between wild-type and *EnCre;Nolz1^{fl/fl}* mutant embryos. We have changed the text as follows: Furthermore, the innervation of the prefrontal cortex was normally initiated in E18.5 *EnCre;Nolz1^{fl/fl}* mutant embryos,*

Point 8: The authors show images at different levels throughout the hypothalamus but the images are truncated at the bottom, so it is not clear if some axons are actually crossing. They should provide the full images.

*Some of the images were indeed truncated at the bottom. We have replaced those images and the panel now clearly shows that there is no midline crossing of DA axons in the hypothalamus of *FoxD1Cre;Nolz1^{fl/fl}* and *En1Cre;Nolz1^{fl/fl}* mutant embryos.*

Minor point 5: check punctuation P9 : ... specification of striatonigral projection neurons and in the absence of *Nolz1* expression several striatopallidal markers are ectopically expressed in the striatum (.)

We have changed this.